# Revealing isoelectronic size conversion dynamics of metal nanoclusters by a noncrystallization approach

Qiaofeng Yao [1], Victor Fung[2], Cheng Sun[1], Sida Huang[1], Tiankai Chen[1], De-en Jiang [2] Jim Yang Lee[1] & Jianping Xie [1]

Atom-by-atom engineering of nanomaterials requires atomic-level knowledge of the size evolution mechanism of nanoparticles, which remains one of the greatest mysteries in nanochemistry. Here we reveal atomic-level dynamics of size evolution reaction of molecular-like nanoparticles, i.e., nanoclusters (NCs) by delicate mass spectrometry (MS) analyses. The model size-conversion reaction is $[Au_{23}(SR)_{16}]^- \rightarrow [Au_{25}(SR)_{18}]^-$ (SR = thiolate ligand). We demonstrate that such isoelectronic (valence electron count is 8 in both NCs) size-conversion occurs by a surface-motif-exchange-induced symmetry-breaking core structure transformation mechanism, surfacing as a definitive reaction of $[Au_{23}(SR)_{16}]^- + 2 [Au_2(SR)_3]^- \rightarrow [Au_{25}(SR)_{18}]^- + 2 [Au(SR)_2]^-$. The detailed tandem MS analyses further suggest the bond susceptibility hierarchies in feed and final Au NCs, shedding mechanistic light on cluster reaction dynamics at atomic level. The MS-based mechanistic approach developed in this study also opens a complementary avenue to X-ray crystallography to reveal size evolution kinetics and dynamics.

[1] Department of Chemical and Biomolecular Engineering, National University of Singapore, 4 Engineering Drive 4, Singapore 117585, Singapore. [2] Department of Chemistry, University of California, Riverside, California 92521, USA. Correspondence and requests for materials should be addressed to J.X. (email: chexiej@nus.edu.sg)

Manipulating functional materials at atom-by-atom basis represents the most ambitious dream of scientists in nanochemistry[1–6]. Such dream should be rooted in precise understanding on growth mechanism of nanoparticles (NPs) at atomic level, which has puzzled nanoscience research community for decades, but little has been unraveled so far[7, 8]. Fundamental breakthrough on particle growth mechanism may come with the recently discovered molecular-like metal NPs. Molecular-like metal NPs or nanoclusters (NCs) possess ultrasmall core diameter ( < 3 nm)[9–12] and feature intriguing molecular-like properties (e.g., single electron transition[13–15], quantized charging[16], intrinsic chirality[17, 18], strong luminescence[19–22], and high catalytic activity and selectivity[23, 24]), which are distinctly different from the metallic behavior (e.g., surface plasmon resonance or SPR) of their larger counterparts (NPs > 3 nm). The molecular-like properties of metal NCs provide sensitive probes for the sizes of metal NCs. In addition, thanks to recent development in cluster synthetic chemistry, metal NCs could now be produced and characterized at atomic precision[25]. Therefore, atomically precise metal NCs could provide ideal platform to reveal the growth mechanisms of ultrasmall metal NPs. In turn, the unraveled size evolution manner of metal NCs could also provide essential clues to the transition (and origin) from molecular NCs to metallic NPs[26, 27].

The continuous efforts devoted to cluster chemistry in the past two decades have produced a considerable number of atomically precise thiolate-protected metal NCs, which can be referred to as $[M_n(SR)_m]^q$, where $n$, $m$, and $q$ are numbers of metal atoms (M), thiolate ligands (SR), and net charges per cluster; in a wide size spectrum, such as $n = 10–329$ for Au NCs[28–33], 7–374 for Ag NCs[34–39], and 18–329 for alloy AuAg NCs[40–42]. X-ray crystallography examinations also revealed that these metal NCs exist in different core symmetry (e.g., face centered cubic (FCC), icosahedral, and dodecahedral) and valence electron counts ($N^\star = n–m–q$)[29, 34, 36–38, 43–47]. Based on the known NC structures and some advanced time-evolution composition/structure monitoring techniques, the size evolution mechanism of metal NCs has been investigated in a couple of contributions with molecular resolution, where the molecular-like reaction equations were proposed for the evolution of $[Au_n(SR)_m]^q$. For example, we recently mapped out balanced reactions for the formation of a series of icosahedron-based NCs (i.e., $[Au_{25}(SR)_{18}]^-$, $[Au_{38}(SR)_{24}]^0$, and $[Au_{44}(SR)_{26}]^{2-}$), and demonstrated that the nucleation and growth of metal NCs are dominantly prompted by stepwise (with a pace of two electrons) boosting of valence electron counts (i.e., growing Au(0) core) under reductive environment[48, 49]. In addition to valence electron-count-elevating size growth, Zeng et al.[50] monitored a valence electron-count-degrading size conversion by matrix-assisted laser desorption/ionization mass spectrometry (MS). Their data suggest that the icosahedron-based $[Au_{38}(SR)_{24}]^0$ ($N^\star = 14$) could be converted into FCC $[Au_{36}(SR')_{24}]^0$ ($N^\star = 12$, SR' denotes a different thiolate ligand from SR) via a ligand-replacement-induced disproportionation reaction. Remarkable molecular-level advances on size evolution mechanism could also be found in recent experimental and theoretical contributions from other groups[51–55]. Despite the aforementioned molecular-level understandings, the size-conversion mechanism of metal NCs has rarely been explored at atomic level, especially for those metal NCs capped with unvaried thiolate ligands. This is most probably due to a lack of good techniques to precisely probe the composition and structure changes of clusters at atomic level.

Here we exemplify that, beyond definitive molecular-like reaction equations, the atomic-level insights on size-conversion reaction dynamics (i.e., behavior of individual atom) of metal NCs could be revealed by a noncrystallization approach based on

systematic electrospray ionization MS (ESI-MS) and tandem MS (MS/MS) analyses. The model size-conversion reaction is a symmetry-breaking isoelectronic growth from $[Au_{23}(SR)_{16}]^-$ ($N^\star = 8$) to $[Au_{25}(SR)_{18}]^-$ ($N^\star = 8$). Complementary to X-ray crystallography analysis, here steady state, time-course, and MS/MS are used to monitor the size conversion in the presence of Au(I)-SR complexes with identical and varied SR ligands. Based on the as-obtained MS data and comprehensive kinetic analyses, we are able to surface a governing reaction pathway for the titled size-conversion reaction with atomic resolution. This size-conversion reaction is initiated by adsorption of two molecules of $[Au_2(SR)_3]^-$ on $[Au_{23}(SR)_{16}]^-$, and simultaneously releases two molecules of $[Au(SR)_2]^-$. Such surface-motif-exchange (SME) reaction could then induce core structure transformation of Au NCs from cuboctahedron (featured by $Au_{13}$ core of $[Au_{23}(SR)_{16}]^-$)[56] to icosahedron (featured by $Au_{13}$ core of $[Au_{25}(SR)_{18}]^-$)[13]. The as-revealed SME-induced core structure transformation mechanism highlights the pivotal impact of the surface protecting motifs on the core structure, shedding mechanistic light on the symmetry-breaking core structure evolution of metal NCs. The atomic-level understanding on size-conversion reaction could also be used to develop efficient methods to stabilize metastable $[Au_{23}(SR)_{16}]^-$ in solution. The development of MS-based mechanistic approach opens a complementary avenue to X-ray crystallography to reveal size evolution kinetics and dynamics, which will be particularly beneficial for the fundamental understandings of water-soluble metal NCs as the crystallization of such metal NCs is still challenging.

## Results

**Isoelectronic size conversion from Au$_{23}$ to Au$_{25}$ NCs**. The model size-conversion reaction used in this study is isoelectronic conversion (which is less explored in comparison to those $N^\star$-elevating or degrading reactions) of water-soluble 8 e$^-$ Au NCs, i.e., $[Au_{23}(p\text{-MBA})_{16}]^- \rightarrow [Au_{25}(p\text{-MBA})_{18}]^-$ (Fig. 1a, $p$-MBA denotes $para$-mercaptobenzoic acid). Such isoelectronic conversion was originally observed as a step reaction in the formation[49] and ligand exchange[57] of $[Au_{25}(SR)_{18}]^-$, and it was chosen in this study due to its short reaction route (consisting of one or two elementary reactions, vide infra). The synthesis of $[Au_{23}(p\text{-MBA})_{16}]^-$ was conducted by a carbon monoxide (CO)-mediated reduction method, where Au(I)-($p$-MBA) complexes were reduced by CO at pH 12.3 in a mixed solvent of water/ethanol (6/4, Vol/Vol; see Methods for more details). The as-prepared $[Au_{23}(SR)_{16}]^-$ NCs are dark green in solution (Fig. 1b, inset) and exhibit absorption features at 589 nm (peak) and 470 nm (shoulder) in its ultraviolet-visible (UV-vis) absorption spectrum (Fig. 1b). This absorption spectrum is in good accordance with that of the reported organic-soluble $[Au_{23}(SR)_{16}]^-$, except for a slight red shift of the absorption peaks[56]. This shift is attributed to the aromaticity and solvation of $p$-MBA ligands on the Au NC surface[58].

To unambiguously determine cluster formula, we performed ESI-MS (in negative ion mode). The as-prepared Au NCs were purified by ultrafiltration and redissolved in water/ethanol (6/4, Vol/Vol) before ESI-MS examination. Three sets of peaks were observed in a broad m/z range of 1000–4000 in their ESI mass spectrum (Fig. 1d, top spectrum), where the most prominent two sets of peaks correspond to the cluster peaks of Au$_{23}$($p$-MBA)$_{16}$ with 5 (centered at m/z = 1430) and 6 (centered at m/z = 1188) negative charges. The good accuracy of our assignment is confirmed by the zoom-in mass spectrum (Fig. 1d, middle spectrum) and isotope analysis of $[Au_{23}(p\text{-MBA})_{16} + 8 \text{ Na} - 12 \text{ H}]^{5-}$ (Fig. 1d, bottom spectrum). The asterisk set of peaks centered at m/z = 1320 could be assigned to $[Au_{22}(SR)_{14}]^0$ (Supplementary

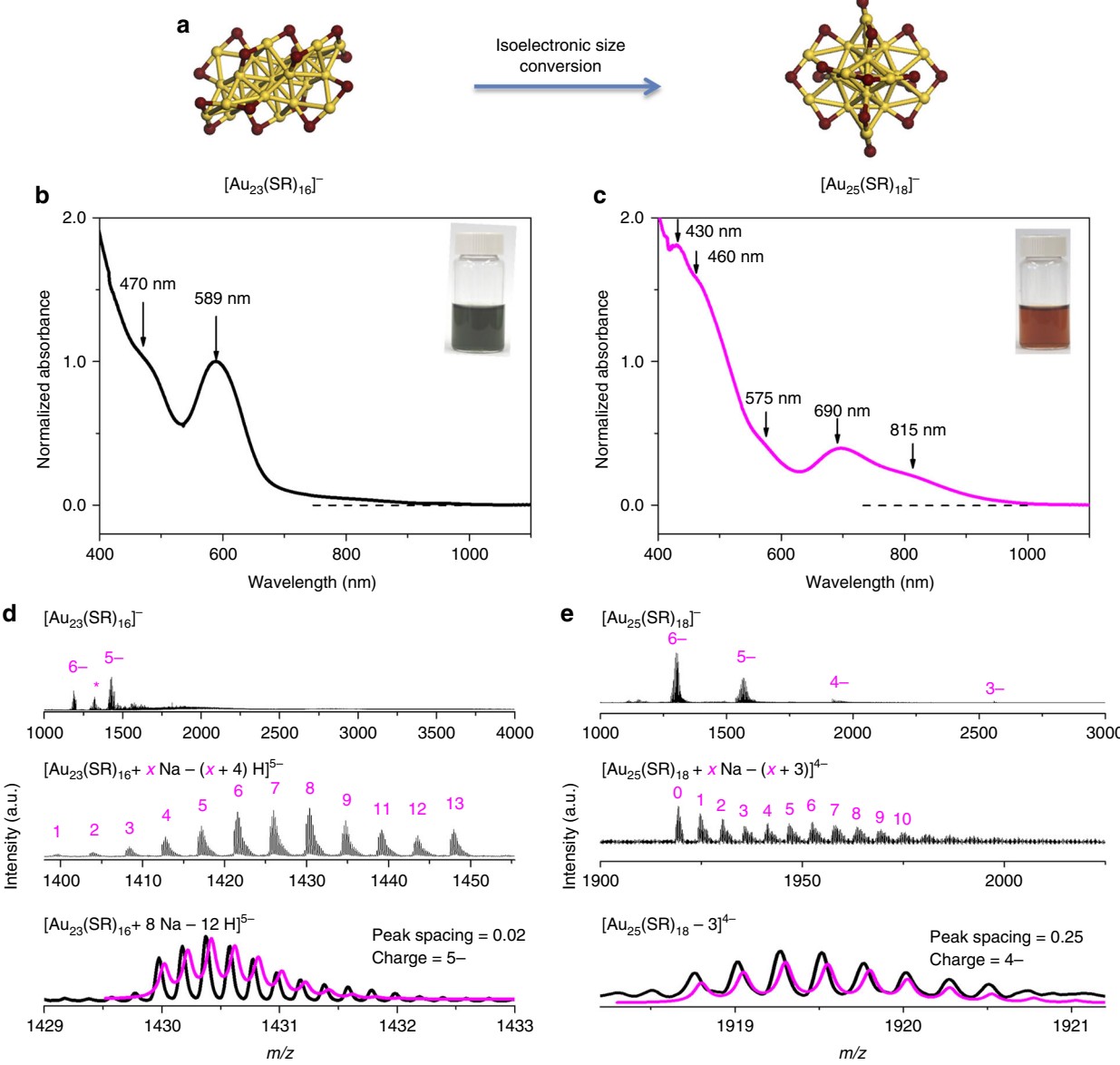

**Fig. 1** Size conversion from $Au_{23}$ to $Au_{25}$ nanoclusters. **a** Schematic illustration of size-conversion reaction from $[Au_{23}(SR)_{16}]^-$ to $[Au_{25}(SR)_{18}]^-$ (yellow, Au; wine, S), where SR denotes thiolate ligand. **b,c** Ultraviolet-visible absorption and **d,e** electrospray ionization mass spectra of **b,d** $[Au_{23}(SR)_{16}]^-$ and **c, e** $[Au_{25}(SR)_{18}]^-$. The crystal structures of $[Au_{23}(SR)_{16}]^-$ and $[Au_{25}(SR)_{18}]^-$ are drawn according to the reported $Au_{23}S_{16}$[56] and $Au_{25}S_{18}$[13] skeletons, where all hydrocarbon tails are omitted for clarity. Insets in **b,c** are digital photos of the corresponding cluster solution. The magenta lines in **d,e** show simulated isotope patterns of the labeled cluster formulas, which match perfectly with the corresponding experimental data

Fig. 1), which is a fragment of $[Au_{23}(p\text{-MBA})_{16}]^-$ generated during the MS measurement (see detailed fragmentation analyses of $[Au_{23}(p\text{-MBA})_{16}]^-$ in the sub-section "Size-conversion dynamics at atomic level" below). It should be noted that such interference from fragments in semi-aqueous medium could be eliminated by transferring the as-obtained Au NCs into organic phase prior to ESI-MS examination, corroborating good molecular purity of $[Au_{23}(p\text{-MBA})_{16}]^-$ in our sample (Supplementary Fig. 2 and 3, and Supplementary Note 1).

The susceptibility of $[Au_{23}(p\text{-MBA})_{16}]^-$ NCs in aqueous or semi-aqueous solution further motivated us to explore the size-conversion reaction of $[Au_{23}(p\text{-MBA})_{16}]^-$. The size-conversion reaction could be simply induced by changing the solvent polarity (e.g., from water/ethanol to pure water). In particular, the purified $[Au_{23}(p\text{-MBA})_{16}]^-$ NCs were dissolved in water, followed by

incubation in a shaker (600 r.p.m., 25 °C) for 2 days. Such incubation would change color of the aqueous Au NC solution from dark green to reddish brown (Fig. 1c, inset). Associated with the distinct color change is the difference in their UV-vis absorption spectrum (Fig. 1c), which shows characteristic absorption peaks of $[Au_{25}(p\text{-MBA})_{18}]^-$ at 815, 690, 575, 460, and 430 nm[49]. ESI-MS analysis on the incubated Au NCs also produces a clean profile of $[Au_{25}(p\text{-MBA})_{18}]^-$ (Fig. 1e). The combined ESI-MS and UV-vis absorption examinations suggest a complete size conversion from $[Au_{23}(SR)_{16}]^-$ to $[Au_{25}(SR)_{18}]^-$ under different solvent polarity. The effectiveness of solvent polarity on cluster conversion was further validated by comparing the UV-vis absorption spectra of the Au NCs in a simulated mother liquid (water/ethanol, 6/4, Vol/Vol) and those in water, where a similar incubation for 2 days only led to minor changes

in the absorption spectra of the former (Supplementary Fig. 4). The polarity-induced size conversion could be attributed to the solvent-dependent stability of SR-[Au(I)-SR]$_n$ protecting motifs. It has been known that less polar solvents would prefer to accommodate longer water-soluble SR-[Au(I)-SR]$_n$ complexes or motifs[22]. By elevating the polarity of the medium (e.g., changing the solvent from water/ethanol to pure water), the long SR-[Au(I)-SR]$_3$ protecting motif of [Au$_{23}$(SR)$_{16}$]$^-$ would become less thermodynamically favored[56]. As an attempt to minimize the total energy, such long motifs would be tailored into shorter analogs, of which the SR-[Au(I)-SR]$_2$ motif featured in the protecting shell of [Au$_{25}$(SR)$_{18}$]$^-$ is a good candidate[13]. Therefore, it is reasonable to anticipate that the size conversion from [Au$_{23}$(SR)$_{16}$]$^-$ to [Au$_{25}$(SR)$_{18}$]$^-$ is thermodynamically driven by the solvent affinity of SR-[Au(I)-SR]$_n$ protecting motifs.

**Size-conversion kinetics at molecular level.** To gain insights into the underlying chemistry of the size-conversion reaction, we first tracked the reaction by time-course UV-vis absorption and ESI-MS analyses. Figure 2a depicts time-course UV-vis absorption spectra of the size-conversion reaction, where characteristic absorptions of [Au$_{25}$(SR)$_{18}$]$^-$ (e.g., at 690 nm) become prominent, while those of [Au$_{23}$(SR)$_{16}$]$^-$ (e.g., at 589 nm) diminish. The

time-course UV-vis absorption data is in good accordance with the distinct color change of the reaction solution from dark green, to brownish green, brown, and finally to reddish brown (insets of Fig. 2a), which unambiguously suggest the formation of [Au$_{25}$(SR)$_{18}$]$^-$ at the expense of [Au$_{23}$(SR)$_{16}$]$^-$. Of particular note, an isosbestic point was observed at ~ 660 nm in the time-course UV-vis absorption spectra, which implies one-to-one conversion stoichiometry from [Au$_{23}$(SR)$_{16}$]$^-$ to [Au$_{25}$(SR)$_{18}$]$^-$. The time-course ESI mass spectra (Fig. 2b) are also supportive to the one-to-one size-conversion mechanism, where the enhancement of [Au$_{25}$(SR)$_{18}$]$^-$ cluster peaks is accompanied by the decline of [Au$_{23}$(SR)$_{16}$]$^-$ cluster peaks.

By noting the identical valence electron count ($N^* = 8$) in both clusters, we were strongly encouraged to explore the redox-dependence of reaction kinetics. The size-conversion reaction was allowed to occur in water pre-saturated by reductive (CO), inert (N$_2$), or oxidative (O$_2$) gas, whereas the reaction kinetics was monitored by time-course UV-vis absorption spectrometry (Supplementary Fig. 5a–5d). The decay profile of [Au$_{23}$(SR)$_{16}$]$^-$ could be fitted by a pseudo-first-order rate equation (Supplementary Fig. 5e–5h). As illustrated in Fig. 2c (entries 1–4, from left), the as-deduced rate constant $k$ is irrelevant to the reducing/oxidizing power of the reaction solution. The irrelevance of rate

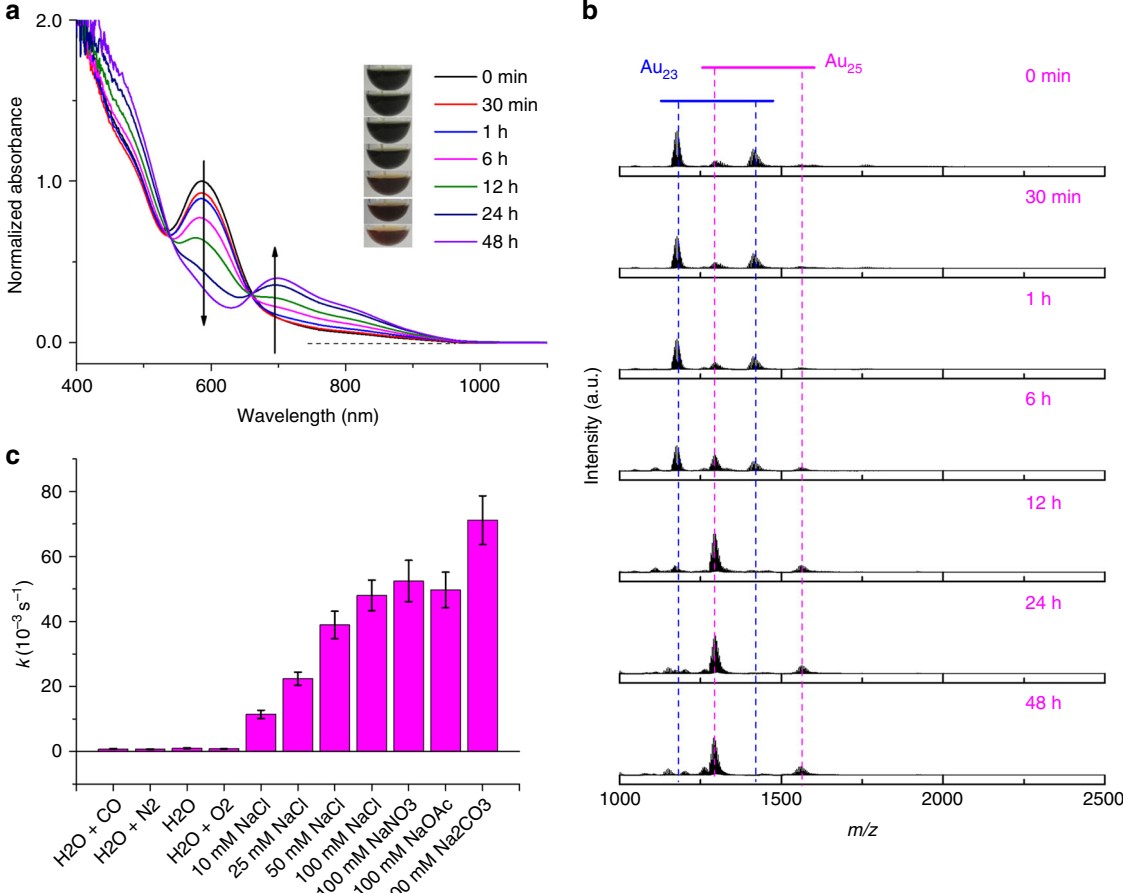

**Fig. 2** Size-conversion kinetics from Au$_{23}$ to Au$_{25}$ nanoclusters. Time-course **a** ultraviolet-visible absorption and **b** electrospray ionization mass spectra of isoelectronic size conversion from [Au$_{23}$(SR)$_{16}$]$^-$ to [Au$_{25}$(SR)$_{18}$]$^-$ in water, where SR denotes thiolate ligand. **c** Rate constant ($k$) of the size-conversion reaction occurred in varied solutions, where OAc denotes acetate. The insets in **a** are digital photos of the reaction mixtures taken at varied reaction time. All absorption spectra are normalized to optical density at 589 nm at $t = 0$. The $k$ values in **c** are deduced according to the characteristic absorption decay profile of [Au$_{23}$(SR)$_{16}$]$^-$ by a pseudo-first-order reaction equation OD$_{@589}$ = OD$_{@589,0}$ × e$^{-kt}$ + $b$, where OD$_{@589,0}$ and OD$_{@589}$ are normalized initial and time-dependent optical density at 589 nm, respectively, whereas $b$ is a constant accounting for the non-zero absorbance at 589 nm at the end state of reaction; the error bars indicate the SEs of the regression

constant towards reducing/oxidizing power agrees well with the isoelectronic nature (8 e⁻ → 8 e⁻) of the size-conversion reaction presented in this study. Of note, this finding is however in sharp contrast to the recently reported reduction-induced reverse size conversion from $[Au_{25}(SeR)_{18}]^-$ to $[Au_{23}(SeR)_{16}]^-$ (SeR = selenolate ligand), where a reduction process is indispensible to activate the stable $[Au_{25}(SeR)_{18}]^-$ in solution[57]. Taking the irrelevance to reducing/oxidizing power, together with one-to-one size-conversion stoichiometry and an apparent $Au_2SR_2$ composition difference between $[Au_{25}(SR)_{18}]^-$ and $[Au_{23}(SR)_{16}]^-$ into consideration, we rationalized that the size conversion should occur by adding 0 e⁻ Au(I)-SR complexes to $[Au_{23}(SR)_{16}]^-$, forming $[Au_{25}(SR)_{18}]^-$ as the final product.

Such Au(I)-SR complex-associated size-conversion mechanism was first supported by the presence of residual Au(I)-SR complex species in the as-prepared $[Au_{23}(SR)_{16}]^-$ solution. We examined the low m/z regime of ESI mass spectrum of the aqueous solution of $[Au_{23}(SR)_{16}]^-$ employed in the size-conversion reaction. A couple of Au(I)-SR complexes with or without intrinsic charge of 1- are identified as $[Au(SR)_2]^-$, $[Au_2(SR)_3]^-$, $[Au_3(SR)_4]^-$, $[Au_4(SR)_4]^0$, and $[Au_4(SR)_5]^-$ (Supplementary Fig. 6). Further insights into the chemical identity of the incoming Au(I)-SR complexes could be generated from the kinetics analyses in reaction solutions of various ionic strength ($I$). As shown in Fig. 2c (entries 5–11), size-conversion reaction exhibits accelerated kinetics with elevated ionic strength, but almost unaltered kinetics towards changed chemical identities of salt. More details about such kinetic analyses are included in Supplementary Fig. 7, 8, and Supplementary Note 2. The as-demonstrated ionic strength dependence should be attributed to electrostatic repulsion between the similarly negatively charged $[Au_{23}(SR)_{16}]^-$ and Au(I)-SR complex species, of which $[Au(SR)_2]^-$, $[Au_2(SR)_3]^-$, $[Au_3(SR)_4]^-$, and $[Au_4(SR)_5]^-$ could be the candidates. At high ionic strength, the squeezed double layer of $[Au_{23}(SR)_{16}]^-$ could weaken its electrostatic repulsion with

negatively charged Au(I)-SR complex species, facilitating their collision and thus size-conversion reaction.

To unambiguously identify reaction equation governing the size conversion from $[Au_{23}(SR)_{16}]^-$ to $[Au_{25}(SR)_{18}]^-$, we performed size-conversion reaction of $[Au_{23}(SR)_{16}]^-$ in the presence of foreign Au(I)-SR' complexes, where SR' denotes a thiolate ligand different from SR. In particular, Au(I)-$p$-NTP complexes ($p$-NTP denotes $para$-nitrothiophenol) were first prepared by mixing $HAuCl_4$ with $p$-NTP. The complexes were then reacted with $[Au_{23}(p\text{-}MBA)_{16}]^-$ in water at varied dosages (expressed by the concentration of Au(I) species, [Au(I)]). As illustrated in Fig. 3a-h, the reactions between $[Au_{23}(p\text{-}MBA)_{16}]^-$ and Au(I)-($p$-NTP) complexes could consistently lead to the formation of $Au_{25}$ NCs in solution. However, a careful comparison of the isotope patterns of NC ions obtained at different Au(I)-($p$-NTP) dosages (Fig. 3i-l), which should be attributed to the molecular weight (MW) difference of $p$-MBA (MW = 153.19) and $p$-NTP (MW = 154.17) ligands. The most intriguing finding is stepwise (with a pace of 3 Da) mass shift when the dosage of Au(I)-($p$-NTP) complexes is increased (Supplementary Fig. 9 and Supplementary Note 3). Remarkably, in comparison with the reference $[Au_{25}(p\text{-}MBA)_{18}]^-$ peak (Fig. 3a,e and i), the most intensive isotope peak exhibits a mass increment of initially 6 Da (Fig. 3b,f and j) and thereafter 3 Da (Fig. 3c,d,g,h,k and l), corresponding to substitutions of 6 and 3 $p$-MBA by $p$-NTP, respectively. Considering similar linear configuration (i.e., SR-[Au(I)-SR]$_n$) of Au(I)-SR complexes and Au(I)-SR protecting motifs, together with the centrosymmetry of $[Au_{23}(SR)_{16}]^-$, such stepwise ligand substitution suggests a SR-[Au(I)-SR]$_2$ motif-based association mechanism. As a dominant pathway, the size-conversion reaction of $Au_{23}$ to $Au_{25}$ NCs is induced by association of two molecules of ($p$-NTP)-[Au(I)-($p$-NTP)]$_2$ with $[Au_{23}(p\text{-}MBA)_{16}]^-$ at a low dosage of complexes ([Au(I)] = 0.1 mM). In accompany with the association of ($p$-NTP)-[Au(I)-($p$-NTP)]$_2$ is dissociation of two molecules of ($p$-MBA)-Au(I)-($p$-MBA), giving rise to a balanced

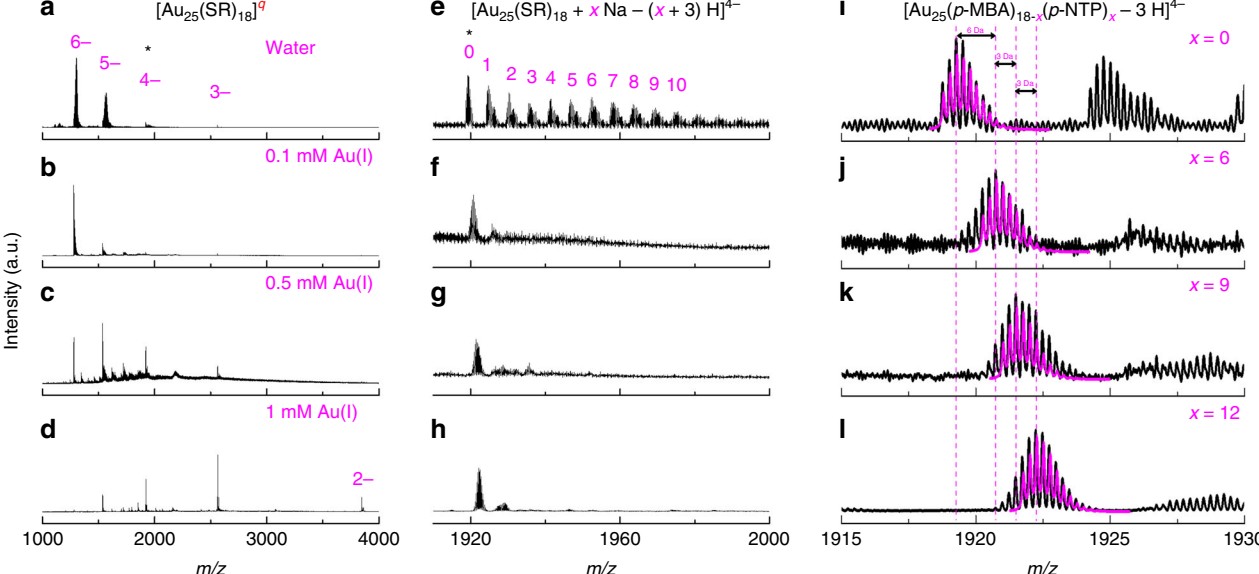

**Fig. 3** Size-conversion reaction induced by foreign Au(I)-SR' complexes. Electrospray ionization mass spectra of $[Au_{25}(SR)_{18}]^-$ nanoclusters formed by reacting $[Au_{23}(p\text{-}MBA)_{16}]^-$ with varied dosage of Au(I)-($p$-NTP) complexes, which are expressed by [Au(I)] = **a,e,i** 0 mM; **b,f,j** 0.1 mM; **c,g,k** 0.5 mM; **d,h,l** 1.0 mM. **a–d** Broad range spectra at m/z = 1000–4000; **e–h** zoom-in spectra of cluster peaks carrying 4- charges; and **i–l** experimental (black lines) and simulated (magenta lines) isotope patterns of $[Au_{25}(p\text{-}MBA)_{18-x}(p\text{-}NTP)_x - 3\,H]^{4-}$ with $x$ values indicated in the corresponding panels. SR and SR' denote different thiolate ligands, whereas $p$-MBA and $p$-NTP are $para$-mercaptobenzoic acid and $para$-nitrothiophenol, respectively

size-conversion reaction, as shown in equation (1). After the size-conversion reaction (equation (1)) is the motif exchange reaction at higher dosages of Au(I)-(p-NTP) complexes ([Au(I)] = 0.5 mM and 1.0 mM), where the excess (p-NTP)-[Au(I)-(p-NTP)]$_2$ complexes could induce stepwise surface exchange reaction with the (p-MBA)-[Au(I)-(p-MBA)]$_2$ protecting motif of Au$_{25}$ NCs (equation (2)).

$$\left[Au_{23}(SR)_{16}\right]^- + 2\left[Au_2(SR')_3\right]^- \rightarrow$$
$$\left[Au_{25}(SR)_{12}(SR')_6\right]^- + 2\left[Au(SR)_2\right]^- \quad (1)$$

$$\left[Au_{25}(SR)_{18-3x}(SR')_{3x}\right]^- + y\left[Au_2(SR')_3\right]^- \rightarrow$$
$$\left[Au_{25}(SR)_{18-3(x+y)}(SR')_{3(x+y)}\right]^- + y\left[Au_2(SR)_3\right]^- \quad (2)$$

To evaluate the magnitude of the thermodynamic driving force for equation (1), we computed its reaction energy by density functional theory (DFT) at the TPSS/def2-SV(P) level with an implicit solvation model. Such energy was found to be −17.5 kcal per mol. Given the same number of similar species before and after the reaction, we expect that the entropy change of equation (1) would be close to zero, so our DFT computation confirms that equation (1) is indeed a thermodynamically favorable reaction. To map out the energetics of the complete conversion pathway at the atomic level is still computationally challenging. Below we propose a mechanistic pathway based on the known structures of [Au$_{23}$(SR)$_{16}$]$^-$ and [Au$_{25}$(SR)$_{18}$]$^-$.

**Size-conversion dynamics at atomic level.** Understanding the size-conversion dynamics at atomic level is of center interest in this study. To gain such atomic-level understanding, we first examined structural similarity/difference of [Au$_{23}$(SR)$_{16}$]$^-$ and [Au$_{25}$(SR)$_{18}$]$^-$ (Fig. 4). The documented X-ray crystallography analyses reveal that both clusters consist of Au$_{13}$ cores, which are organized into center-occupied cuboctahedron (inset of Fig. 4a) and icosahedron (inset of Fig. 4f) in [Au$_{23}$(SR)$_{16}$]$^-$ and [Au$_{25}$(SR)$_{18}$]$^-$, respectively[13, 56]. The icosahedral Au$_{13}$ core of [Au$_{25}$(SR)$_{18}$]$^-$ is protected by six dimeric SR-[Au(I)-SR]$_2$ motifs (Fig. 4f), whereas the cuboctahedral Au$_{13}$ core of [Au$_{23}$(SR)$_{16}$]$^-$ is wrapped by two trimeric SR-[Au(I)-SR]$_3$ motifs, two monomeric SR-Au(I)-SR motifs, and four bridging SR (Fig. 4a). Despite the abovementioned SR-[Au(I)-SR]$_n$ motifs, two unique Au atoms are present in [Au$_{23}$(SR)$_{16}$]$^-$ (highlighted in blue in Fig. 4), which act as hubs connecting trimeric and monomeric protecting motifs.

Based on the crystal structures of these two clusters, as well as the reaction equation depicted in equation (1), we are now able to propose a reasonable reaction dynamics for the size-conversion reaction. As schematically illustrated in Fig. 4a,b, the size conversion is induced by association of two SR-[Au(I)-SR]$_2$ motifs with [Au$_{23}$(SR)$_{16}$]$^-$, most probably with two Au atoms in each motif capped on one square and one triangular facets of cuboctahedral core. It should be pointed out that only the front view of [Au$_{23}$(SR)$_{16}$]$^-$ (showing association of one SR-[Au(I)-SR]$_2$ motif) was depicted in Fig. 4b, and the association of the other SR-[Au(I)-SR]$_2$ motif would occur in a similar way at the centrosymmetric back sites. The steric hindrance effects and/or the electrostatic repulsion between the incoming SR-[Au(I)-SR]$_2$ motifs and the neighbored bridging SR (indicated by

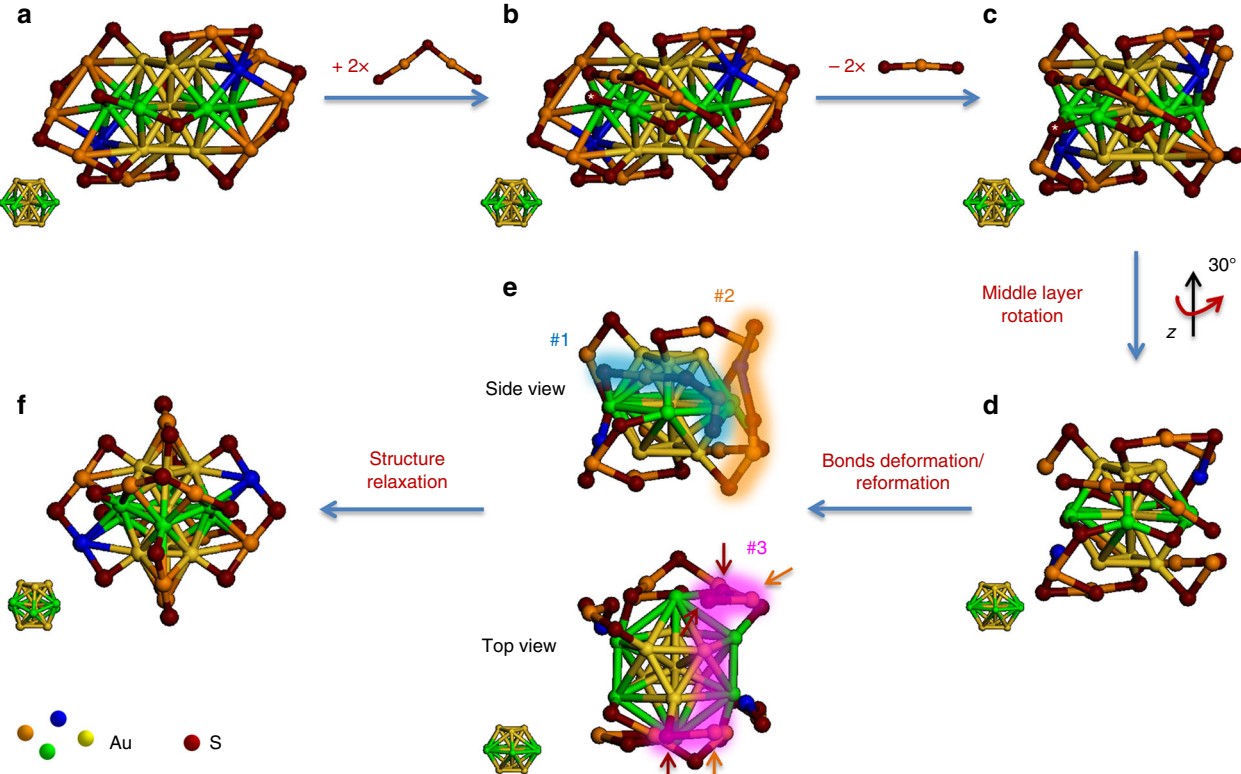

**Fig. 4** Schematic illustration of size-conversion reaction dynamics. **a** Structure of [Au$_{23}$(SR)$_{16}$]$^-$, **b** association of 2 SR-[Au(I)-SR]$_2$, **c** dissociation of 2 SR-Au(I)-SR, **d** core structure transformation, **e** surface rearrangement, and **f** structure relaxation to [Au$_{25}$(SR)$_{18}$]$^-$. SR denotes thiolate ligand. The hub atoms and middle layer of Au$_{13}$ core in [Au$_{23}$(SR)$_{16}$]$^-$ are highlighted in blue and green, respectively, for easy identification. The formation of dimeric SR-[Au(I)-SR]$_2$ motifs #1–#3 is shown by glowing belts in **e**, and the S atoms and Au atoms involved in the formation of such dimeric motif #3 are indicated by wine and orange arrows, respectively. The core structure at each step is shown at left bottom of each panel

asterisk S in Fig. 4b) would then push such bridging SR downward, approaching to a Au atom of SR-[Au(I)-SR]$_3$ motif that sits close to the asterisk SR. Such approaching could disturb the Au-S bonding in SR-[Au(I)-SR]$_3$ motif, leading to ejection of the middle SR-Au(I)-SR module as well as bond formation between the remaining Au atom and the asterisk S (Fig. 4c).

It should be noted that the cuboctahedral Au$_{13}$ core of [Au$_{23}$(SR)$_{16}$]$^-$ constitutes three layers of vertex Au atoms (i.e., Au$_3$-Au$_6$-Au$_3$) and a center Au atom. As only trimeric SR-[Au(I)-SR]$_3$ motifs coordinate to the Au atoms from all these three layers, they are essential to stabilize the cuboctahedral configuration of Au$_{13}$ core. This assertion is well supported by the ligand exchange-induced size-transformation reactions documented in the literature, where the cuboctahedral or quasi-FCC symmetry is retained in the metal core of product NCs as long as the trimeric SR-[Au(I)-SR]$_3$ motifs are kept complete[59, 60]. The destruction of SR-[Au(I)-SR]$_3$ motifs, induced by association of SR-[Au(I)-SR]$_2$ could, on the other hand, enables Au$_{13}$ core to undergo a symmetry-breaking configuration transformation from cuboctahedron to icosahedron. Fig. 4c-f present a plausible dynamics for such transition. The middle layer of cuboctahedron (highlighted in green) rotates 30° according to a C$_3$ axis (arbitrarily assigned as z axis in Fig. 4). Such middle layer rotation could also make the end SR groups of associated SR-[Au(I)-SR]$_2$ approaching to the dangling hub Au atom (Fig. 4d). Based on the as-formed landscape of the surface protecting modules, a bond deformation/reformation process would occur to assemble these surface protecting modules into six SR-[Au(I)-SR]$_2$ motifs. The surface Au atoms and SR ligands involved in the formation of each SR-[Au(I)-SR]$_2$ motif are highlighted by glowing belts in Fig. 4e (only three representative motifs are shown). Accompanying with the bond deformation/reformation is core structure relaxation, which would turn the middle layer of Au$_{13}$ core from a planar hexagonal configuration into a chair-like configuration (Fig. 4f). As a net

result, the SME reaction between two foreign SR-[Au(I)-SR]$_2$ motifs and two pristine SR-Au(I)-SR motifs induces not only core structure transformation from cuboctahedron to a more compact icosahedron, but also surface rearrangement generating six SR-[Au(I)-SR]$_2$ motifs, ultimately giving rise to [Au$_{25}$(SR)$_{18}$]$^-$. We rationalizes that such SME-induced core structure transformation mechanism originates from the robust linear structure of SR-[Au(I)-SR]$_n$ protecting motifs, which is in sharp contrast to the documented ligand exchange-induced size-transformation mechanism[50].

The most reliable way to validate the proposed SME-induced core structure transformation mechanism is resolving crystal structure of all intermediate cluster species by X-ray crystallography. Such crystallization-based approach was initially utilized to probe the exchangeable sites of metal NCs in ligand exchange reactions[61, 62], and more recently this approach was employed by Jin and colleagues[60] to monitoring a ligand exchange-induced symmetry-retaining transformation from [Au$_{23}$(SR)$_{16}$]$^-$ to [Au$_{21}$(SR)$_{12}$(Ph$_2$PCH$_2$PPh$_2$)$_2$]$^+$ NCs. It should be pointed out that all identifiable intermediate species in the aforementioned successful attempts possess an identical M-S framework as that of the corresponding parent metal NCs, which on one hand makes possible co-crystallization of parent and intermediate NCs for X-ray crystallography examination, but on the other hand requires a similar packing symmetry in the feed and product NCs. Due to such fastidious symmetry requirement as well as the tremendous difficulties in crystallization of reactive intermediate cluster species (especially for water-soluble ones), the crystallization-based approach might be less effective in revealing the dynamics of the symmetry-breaking size-conversion reaction presented in this study.

Alternatively, delicate insights on reaction dynamics of presented size conversion could be revealed by MS/MS analysis. In a typical MS/MS analysis, the cluster ions of interest are selected in the primary MS (MS-1) analysis, followed by a

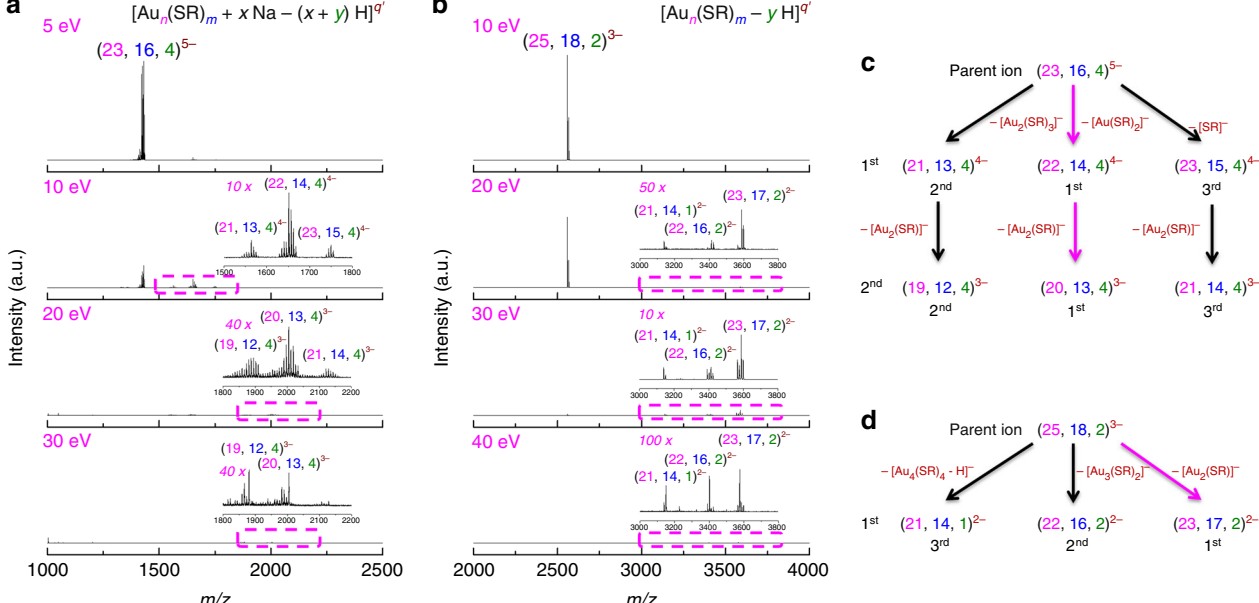

**Fig. 5** Structural relatedness between Au$_{23}$ and Au$_{25}$ nanoclusters. **a,b** Tandem mass spectra and **c,d** fragmentation pathways of **a,c** [Au$_{23}$(SR)$_{16}$]$^-$ and **b,d** [Au$_{25}$(SR)$_{18}$]$^-$. SR denotes thiolate ligand. For clarity purpose, the cluster and fragment cluster ions [Au$_n$(SR)$_m$ + x Na − (x + y) H]$^{q'}$ are referred to as (n, m, y)$^{q'}$. The applied collision energies are indicated at the left-top corner of each spectrum in **a,b**, while the inset in each row shows boxed area in the corresponding spectrum. The abundance order of the fragment cluster ions is summarized underneath the corresponding ions, whereas the dominant fragmentation pathways are highlighted by the magenta arrows

fragmentation analysis in the successive MS (MS-2) analysis. By monitoring the fragmentation preference at varied collision energies, the MS/MS analysis could not only shed light on the structure of biomolecules or molecular-like clusters, but also reveal the bond susceptibility hierarchy within the structures[63–65]. We monitored the fragmentation spectra of quintuple-negatively charged $Au_{23}$ cluster ions with m/z centered at ~ 1430 (Fig. 5). As the cluster anions detected in MS could be expressed in a generic form of $[Au_n(SR)_m + x\,Na - (x+y)\,H]^{q'}$ with intrinsic charge $q = q' + y$, the cluster and fragment peaks in MS/MS analysis will be hereafter referred to as $(n, m, y)^{q'}$ for a clean and clear interpretation. As shown in Fig. 5a, with collision energy increased from 5 to 10 eV, the parent cluster ions $(23, 16, 4)^{5-}$ start to degrade into 1st generation fragment cluster ions $(21, 13, 4)^{4-}$, $(22, 14, 4)^{4-}$, and $(23, 15, 4)^{4-}$. The zoom-in spectrum and isotope patterns of all identified species in Fig. 5 could be found in Supplementary Fig. 10-20, in which the perfect match between the experimental and simulated isotope patterns (with a minor exception in Supplementary Fig. 14 due to the compromised signal-to-noise ratio) validates the good accuracy of our assignment. Of note, all these first-generation fragment cluster ions are quadruple-negatively charged and thus should be formed by dissociation of single-negatively charged $[Au_2(SR)_3]^-$, $[Au(SR)_2]^-$, and $[SR]^-$ from the parent ions, respectively (Fig. 5c). Based on the abundance of first-generation fragment cluster ions in MS (Fig. 5a, 10 eV spectrum; also see the abundance sequence indicated underneath each ion in Fig. 5c), the sequence in departing preference of these small single-negatively charged motifs are $[Au(SR)_2]^- > [Au_2(SR)_3]^- > [SR]^-$. The preferential departure of $[Au(SR)_2]^-$ motif is in good accordance with the proposed SME mechanism, which is also dependent on preferential dissociation of $[Au(SR)_2]^-$ motif.

More supportive evidence to the SME-induced core structure transformation mechanism comes from the comparison of MS/MS spectra of $[Au_{23}(SR)_{16}]^-$ (Fig. 5a) and $[Au_{25}(SR)_{18}]^-$ (Fig. 5b). The proposed SME-induced core structure transformation mechanism suggests that the formation of $[Au_{25}(SR)_{18}]^-$ is heavily dependent on the structure reorganization of $[Au_{23}(SR)_{16}]^-$ after $[Au(SR)_2]^-$ dissociation. This readily implies that some structure features of the intermediate species formed by dissociation of $[Au(SR)_2]^-$ from $[Au_{23}(SR)_{16}]^-$ might be retained in $[Au_{25}(SR)_{18}]^-$. To verify such speculation, we first performed MS/MS analysis on $[Au_{23}(SR)_{16}]^-$ at higher collision energy. As can be seen in Fig. 5a, by elevating collision energy to 20 eV, first-generation fragment cluster ions diminished with emergence of second-generation fragment cluster ions of $(19, 12, 4)^{3-}$, $(20, 13, 4)^{3-}$, and $(21, 14, 4)^{3-}$. A close formula comparison between first- and second-generation fragment cluster ions suggests that the second-generation fragment ions are most probably developed from their first-generation analogs by dissociation of $[Au_2(SR)]^-$ motifs, respectively (Fig. 5c). Such assertion is supported by the unvaried abundance sequence of the corresponding species (those paired by arrows in first and second generations in Fig. 5c) in each generation. As a complementary note, further increasing collision energy (Fig. 5a, 30 eV spectrum) could result in diminishing of second-generation fragment cluster ions, but we did not observe higher generation fragment cluster peaks. This may be due to instability of higher generation fragments.

Of core importance, similar dissociation of $[Au_2(SR)]^-$ motif is also identified as the most prominent fragmentation pathway of $[Au_{25}(SR)_{18}]^-$. The target ions in MS/MS analysis of $Au_{25}$ NCs are $(25, 18, 2)^{3-}$. By elevating the collision energy up to 40 eV, we could only observe first-generation fragment cluster ions in MS/MS spectra (Fig. 5b). The captured first-generation fragment cluster ions are $(21, 14, 1)^{2-}$, $(22, 16, 2)^{2-}$ and $(23, 17, 2)^{2-}$,

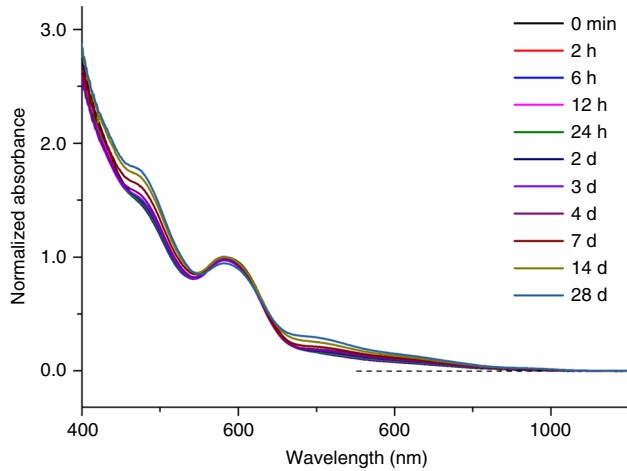

**Fig. 6** Enhanced stability of $[Au_{23}(SR)_{16}@xCTA]^-$. Time-course ultraviolet-visible absorption spectra of $[Au_{23}(SR)_{16}@xCTA]^-$ dissolved in ethanol, where SR and CTA denote thiolate and cetyltrimethylammonium ligands, respectively. All absorption spectra are normalized to optical density at 589 nm at $t = 0$

which are developed by dissociation of $[Au_4(SR)_4 - H]^-$, $[Au_3(SR)_2]^-$, $[Au_2(SR)]^-$, respectively, from the parent ions. The abundance sequence of $(23, 17, 2)^{2-} > (22, 16, 2)^{2-} > (21, 14, 1)^{2-}$ suggests dissociation of $[Au_2(SR)]^-$ as a dominant fragmentation pathway. As a brief sum-up, the dominant fragmentation pathway of $[Au_{23}(SR)_{16}]^-$ is successive dissociation of $[Au(SR)_2]^-$ and $[Au_2(SR)]^-$, respectively, from the parent and first-generation fragment ions (highlighted by magenta arrows in Fig. 5c), whereas the dominant fragmentation pathway of $[Au_{25}(SR)_{18}]^-$ is dissociation of $[Au_2(SR)]^-$ (magenta arrow in Fig. 5d). Such similar fragmentation behavior of the parent ions of $Au_{25}$ NCs and 1st generation fragment cluster ions of $Au_{23}$ NCs implies that they share some common and key structure features, which form the structural basis for the size-conversion reaction and thus providing supportive evidences to the proposed SME-induced core structure transformation mechanism.

**Enhanced stability of $Au_{23}$ NCs by complete protection.** As an important bonus from known reaction kinetics and dynamics, we are able to drive the reaction to selectively produce the reactant or product NCs for both fundamental and practical uses. In comparison with $[Au_{25}(SR)_{18}]^-$, $[Au_{23}(SR)_{16}]^-$ is less explored, most probably due to its less sophisticated synthesis and storage chemistry. By knowing the size-conversion reaction of $[Au_{23}(SR)_{16}]^-$ is induced by association of $[Au_2(SR)_3]^-$ motifs, we hypothesized that the degradation of $[Au_{23}(SR)_{16}]^-$ via size conversion into $[Au_{25}(SR)_{18}]^-$ could be impeded by a better protection of the surface of $[Au_{23}(SR)_{16}]^-$. Such better protecting shell could be established by coating a monolayer of bulky cations on the surface of $[Au_{23}(SR)_{16}]^-$ via a reported phase-transfer assisted ion-paring reaction[66]. In particular, the freshly prepared $[Au_{23}(SR)_{16}]^-$ NCs were allowed to react with an ethanolic solution of cetyltrimethylammonium chloride (CTACl) in the presence of a toluene extraction layer. The ion-pairing reaction between carboxylic groups of p-MBA ligands and $CTA^+$ could anchor a monolayer of CTA on the surface of $[Au_{23}(SR)_{16}]^-$, giving rise to $[Au_{23}(SR)_{16}@xCTA]^-$, which are extractable by organic phase. The as-obtained $[Au_{23}(SR)_{16}@xCTA]^-$ exhibits significantly enhanced solution stability over the parent $[Au_{23}(SR)_{16}]^-$. As shown in the time-course UV-vis absorption spectra (Fig. 6), an elongated incubation of $[Au_{23}(SR)_{16}@xCTA]^-$ for 4 weeks only led to a slight change of their characteristic

absorption peaks, which is in sharp contrast to an accelerated degradation of $[Au_{23}(SR)_{16}]^-$ in either water or the simulated mother liquid (water/ethanol, 6/4, Vol/Vol) within 2 days (Supplementary Fig. 21).

## Discussion

In summary, we have developed a noncrystallization tool box for revealing reaction kinetics and dynamics of a symmetry-breaking size/structure conversion reaction of thiolate-protected Au NCs at atomic level. By a combined use of steady-state and time-dependent UV-vis absorption and ESI-MS techniques to monitor both Au NC and Au(I)-SR complex species involved, we unambiguously demonstrated that the isoelectronic size-conversion reaction of $Au_{23}$ NCs occurred via the definitive equation (1): $[Au_{23}(SR)_{16}]^- + 2\ [Au_2(SR'_3)]^- \rightarrow [Au_{25}(SR)_{12}(SR')_6]^- + 2\ [Au(SR)_2]^-$. Based on such definitive equation, together with structural features reflected by MS/MS analysis, we constructed a SME-induced symmetry-breaking core structure transformation mechanism for the isoelectronic size-conversion reaction, experimentally corroborating the structural relatedness of Au NCs with varied core symmetry. The SME-induced symmetry-breaking core structure transformation mechanism thus not only rationalizes the structure diversity of metal NCs in sub-3 nm regime, but also offers insightful clues towards demystification of long-standing fundamental puzzles, such as nucleation-growth of nanocrystals, molecular-to-metallic transition, and emergence of collective physicochemical properties (e.g., SPR) of metal materials. This work also exemplifies the usefulness of the non-crystallization approach (based on systematic MS and MS/MS investigations) for revealing reaction kinetics and dynamics of cluster reactions with atomic resolution.

## Methods

**Materials**. Hydrogen tetrachloroaurate (III) trihydrate ($HAuCl_4\cdot3H_2O$), cetyl-trimethylammonium bromide (CTABr), $p$-NTP, and $p$-MBA from Sigma Aldrich; sodium hydroxide (NaOH) from Merck; CTACl from Alfa Aesar; ethanol and toluene from Fisher; and carbon monoxide (CO, 99.9%) from Singapore Oxygen Air Liquide Pte Ltd. (SOXAL) were used as-received without further purification. All aqueous solutions were prepared with ultrapure Millipore water (18.2 MΩ·cm). All glassware were washed with aqua regia and repetitively rinsed with ethanol and ultrapure water before use.

**Synthesis of Au$_{23}$ NCs**. $[Au_{23}(p\text{-}MBA)_{16}]^-$ NCs were prepared by a CO-mediated reduction method. In a typical synthesis, 5.5 mL of ultrapure water, 3.5 mL of ethanol, 0.5 mL of 50 mM $p$-MBA ethanolic solution, and 0.25 mL of 50 mM $HAuCl_4$ aqueous solution were added into a 20 mL glass vial in sequence, followed by stirring at 500 r.p.m. for 5 min. At the end of aforementioned mixing process, a light-yellow suspension of Au(I)-($p$-MBA) complexes was formed. Elevating the solution pH to 12.3 by dropwise addition of 1 M NaOH aqueous solution turned such suspension into clear solution. After stirring for another 30 min, CO was bubbled into the reaction mixture for 2 min to initiate reduction of Au(I)-($p$-MBA) complexes. The growth of Au NCs was allowed to proceed air-tightly under mild stirring (500 r.p.m.). After 6 h reaction, a dark green solution was obtained as raw product.

The raw product could be purified by ultrafiltration. In a typical ultrafiltration process, 10 mL of the as-obtained raw NC solution was fed into the processing unit (MW cutoff (MWCO) of 10 kDa). After discarding the filtrate, a cluster concentrate of 10 times of original concentration was recovered. Nine milliliters of water/ethanol (6/4, Vol/Vol) was added into the NC concentrate, which was subjected to another round of ultrafiltration. The NC concentrate of 10 times of the original concentration was recovered for further exploration and characterization.

**Size-conversion reaction from Au$_{23}$ to Au$_{25}$ NCs**. In a typical size-conversion reaction, 0.1 mL of the as-prepared $[Au_{23}(p\text{-}MBA)_{16}]^-$ concentrate was added into 0.9 mL of ultrapure water, followed by incubation in a thermomixer (650 r.p.m., 25 °C). After 2 day incubation, the dark green solution turned into reddish brown, indicating the formation of $[Au_{25}(p\text{-}MBA)_{18}]^-$ NCs.

The size-conversion reaction was also performed in the presence of foreign Au(I)-SR' complexes. In particular, a stock thiolate solution (10 mM) was prepared by dissolving a calculated amount of $p$-NTP into 0.1 M NaOH aqueous solution. Au(I)-($p$-NTP) complex solutions were then prepared by mixing the stock $p$-NTP solution and 10 mM $HAuCl_4$ aqueous solution at a fixed thiol-to-Au ratio of 3:1,

followed by topping up the volume with ultrapure water to designed concentrations. 0.1 mL of $[Au_{23}(p\text{-}MBA)_{16}]^-$ concentrate was added into 0.9 mL of Au(I)-($p$-NTP) complex solution. The reaction mixture was then incubated in a thermomixer (650 r.p.m., 25 °C) for 2 days.

**Phase-transfer of Au$_{23}$ NCs**. Five milliliters of ethanolic solution of 100 mM CTACl or CTABr and 5 mL of toluene were added into an equiv-volume freshly prepared raw $[Au_{23}(p\text{-}MBA)_{16}]^-$ solution in sequence under vigorous stirring (1000 r.p.m.). After stirring for another 2 min, the reaction mixture was kept still for complete phase separation (within ∼ 5 min). Phase-transferred $[Au_{23}(p\text{-}MBA)_{16}@x\text{CTA}]^-$ NCs were collected in the organic phase.

The raw $[Au_{23}(p\text{-}MBA)_{16}@x\text{CTA}]^-$ was then precipitated by centrifugation (14,000 rpm, 5 min) after addition of 4 equiv-volume of hexane to remove excess CTACl (or CTABr) and other impurities. The precipitate was recovered and redissolved in 200 μL of ethanol for further use.

**DFT computation**. Parallel, resolution-of-identity DFT calculations with the TPSS form of the meta generalized gradient approximation for electron exchange and correlation[67] and the def2-SV(P) basis sets were performed with the quantum chemistry program Turbomole V6.5[68]. Effective core potentials which have 19 valence electrons and include scalar relativistic corrections were used for Au[69]. The Conductor-like Screening Model[70] implemented in Turbomole was used to compute the energies of solvated species. $CH_3$ was used to simplify R in $[Au_n(SR)_m]^q$.

**Materials characterization**. Solution pH was monitored by a Mettler Toledo FE 20 pH meter. The size-conversion reactions were carried out in an Eppendorf Comfort thermomixer. UV-vis absorption spectra were obtained on a Shimadzu UV-1800 spectrometer, and the optical intensity was normalized to the characteristic absorption at 589 nm of fresh $[Au_{23}(SR)_{16}]^-$ used in size-conversion exploration unless otherwise indicated. Ultrafiltration was conducted in an Amicon ultrafiltration unit (10 mL) equipped with a membrane filter of MWCO of 10 kDa. ESI-MS and MS/MS were performed on a Bruker microTOF-Q system in negative ion mode, with operating conditions detailed as followings: source temperature 120 °C, dry gas flow rate 8 L per min, nebulizer pressure 3 bar, capillary voltage 3.5 kV, and sample injection rate 3 μL per min.

**Data availability**. All relevant data are available from the corresponding author on request.

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

## Acknowledgements

We thankfully acknowledge the financial support of the Ministry of Education, Singapore (Academic Research Grants R-279-000-481-112 and R-279-000-538-114). The

computational exploration was supported by the Division of Chemical Sciences, Geosciences and Biosciences, Office of Basic Energy Sciences, U.S. Department of Energy.

## Author contributions

J.X. supervised this work. J.X., J.Y.L., and Q.Y. designed the experiments, whereas Q.Y., C.S., S.H., and T.C. carried out the experiments. DFT calculation was conducted by V.F. and D.-e.J. All authors were involved in discussion of results and preparation of manuscript.

## Additional information

**Competing interests:** The authors declare no competing interests.

