## [Peer Review File · Nature Communications]

Reviewers' comments:

Reviewer #1 (Remarks to the Author):

The work by Yao et al. [titled as "Revealing Isoelectronic Size Conversion Dynamics of Metal Nanoclusters by a Noncrystallization Approach"] have demonstrated the dynamics of an isoelectronic size conversion of Au₂₃(pMBA)₁₆ to Au₂₅(pMBA)₁₈ cluster through molecular spectroscopy and mass spectrometric analysis. The overall size conversion happens due to solvent-induced instability. They have further confirmed the mechanism by introducing external thiolates to the Au₂₃ cluster solution. The experimental data supported by DFT calculation to understand how the conversion works. But, I strongly feel that this work lacks the novelty to be considered in journals like Nat. Commun. Following specific reasons make me feel that this work is not suitable for this journal.

1. The authors explain three major novel factors in this manuscript: a. Size conversion b. Isoelectronic conversion, c. Dynamics/kinetics by noncrystallization process;

a. Size conversion: The size conversion of nanoclusters due to instability is very well known; it started from the 1st example of size conversion of Au₁₁ to Au₂₅ (J. Am. Chem. Soc., 2005, 127, 13464–13465). Thereafter many examples are there which explains the size conversion of nanoclusters. From that perspective, this size conversion is not new.

b. Isoelectronic conversion: Jin et al. have recently shown such isoelectronic conversions from Au₂₅ to Au₂₃ for selenolates (Chem. Mater., 2017, 29, 3055–3061, Ref. no 57) by addition of NaBH₄. Then Au₂₃ can be again back to Au₂₅ by introducing thiols (PET) both with and without NaBH₄ case. So, such isoelectronic conversions are already reported even for same clusters, the only differences I can find of is: in this case, its solvent induced whereas for Jin et al.: its ligand and NaBH₄ induced.

c. Dynamics/kinetics by noncrystallization process: The same author group has published many of them (such as J. Am. Chem. Soc., 2014, 136, 10577–10580) for understanding the growth mechanism. Even a new article came just recently in this same journal from the same group on similar work (Nat. Commun. 8, Article number: 927, 2017) where Au₂₅ is converted to Au₄₄ (12e conversion) and the kinetics/dynamics have been studied through noncrystallization process.

Other minor issues:

2. Is that conversion is very specific about ligands? Only p-MBA shows such conversion? This Au₂₃ cluster can also be synthesized with a variety of ligands. Will this solvent induced conversion also work for them?

3. In Figure 2: The author says "that the reaction kinetics is rarely dependent on the chemical identities of anions, but it is highly related to the ionic strength"
If the reaction kinetics is pseudo 1st order then how ionic strength effects in the size conversion rate kinetics? More explanation is needed here.

4. Figure 3: Each stack spectral intensity axis can be narrow down as there is more free space available between each stack. This will help to visualize the spectra in a better way.

5. Ref. 9. does not fit there; a recent important review article on nanoclusters (Chem. Rev., 2017, 117 (12), 8208–8271) should replace ref. 9.

6. In all the absorbance based figures (such as Figure 1 b,c; 2a, etc.), absorbance value is not there. Absorbance should always have some value, it cannot be arbitrary. If you have normalized, then mention normalized absorbance.

7. Many times, the calculated mass spectra and experimental mass spectra do not properly match (such as Figure 3c, x=6,9, 12, SI Figure 2b, SI Figure 3a,) whereas others such as SI Figure 9 to 12 matches perfectly. Anything wrong in the assignment for the earlier case? Figure S13 would be better to consider as noise unless one has better spectra.

Reviewer #2 (Remarks to the Author):

Ligand protected metal clusters have attracted much attention due to their unique physical and chemical properties. Various characterization techniques have progressed for these clusters, much knowledge has been accumulated. One of them is mass spectrometry, which has made it possible to handle metal clusters as molecule or aggregate of atoms. Recently, X-ray crystal structure analysis has become essential techniques to understand synthesized clusters. However, mass spectrometry is also important technique yet. In this work, the authors explored the synthesis mechanism from [Au₂₃(SR)₁₆]- to [Au₂₅(SR)₁₈]- using ESI mass spectrometry. They take advantage of characteristics of mass spectrometry in this work. Additionally, not only many experiments are conducted, but also well analyzed about obtained results. this knowledge and information will fascinate readers of various fields. Therefore, I think this work deserve potential for publication to Nature Communications. To become this work better, I have some minor comments and questions. I am very glad if you will consider about these.

(1) In page 9, the authors explain why the conversion occurs from [Au₂₃(SR)₁₆]- to [Au₂₅(SR)₁₈]- by changing a solvent. If [Au₂₃(SR)₁₆]- prefers to have long oligomer motif, [Au₂₃(SR)₁₆]- seems to decompose instead of conversion to [Au₂₅(SR)₁₈]-? Why did not this cluster decompose?

(2) In page 16, association between 2 SR-[Au(I)SR]₂ and [Au₂₃(SR)₁₆]- is describe. Where does 2 SR-[Au(I)SR]₂ come from?

(3) In page 23, the high stability of [Au₂₃(SR)₁₆@xCTA]- was introduced. But only a time course of this optical absorption spectrum was shown in Figure 6. You might compare this result to that of [Au₂₃(SR)₁₆]- in water/ethanol. By adding this information, reader can understand the difference of stability easily.

(4) In this work, the conversion from [Au₂₃(SR)₁₆]- to [Au₂₅(SR)₁₈]- occurs by changing a solvent from water/ethanol to pure water. Is this reaction reversible?

(5) Experimental isotope patterns in supplementary figure 2 and 3 do not seem to fit calculation data. Is this of calibration problem?

Replies to reviewers' comments and descriptions of revisions made

Comments by Reviewer #1:

The work by Yao et al. [titled as “Revealing Isoelectronic Size Conversion Dynamics of Metal Nanoclusters by a Noncrystallization Approach”] have demonstrated the dynamics of an isoelectronic size conversion of $\text{Au}_{23}(\text{pMBA})_{16}$ to $\text{Au}_{25}(\text{pMBA})_{18}$ cluster through molecular spectroscopy and mass spectrometric analysis. The overall size conversion happens due to solvent-induced instability. They have further confirmed the mechanism by introducing external thiolates to the Au_{23} cluster solution. The experimental data supported by DFT calculation to understand how the conversion works. But, I strongly feel that this work lacks the novelty to be considered in journals like Nat. Commun. Following specific reasons make me feel that this work is not suitable for this journal.

Reply: We are grateful to this reviewer for his/her professional efforts on reviewing our manuscript. In this study, we attempted to reveal the size conversion dynamics from $[\text{Au}_{23}(\text{p-MBA})_{16}]^+$ to $[\text{Au}_{25}(\text{p-MBA})_{18}]^+$ at *atomic level* by combined utilization of optical and mass spectrometry (including tandem mass spectrometry). Atomic-level understandings on the growth mechanism of nanoparticles (NPs) or nanoclusters (NCs) are decisive pre-requisites for customizing nanomaterials at atom-by-atom basis, which is one of the most ambitious dreams of materials scientists. While previous attempts were devoted into revealing size growth mechanism of NCs at molecular level (i.e., evolution sequence of $[\text{Au}_n(\text{SR})_m]^q$ NCs, where n , m and q are numbers of gold atom, thiolate ligand and net charge per cluster, respectively; *J. Am. Chem. Soc.* **2014**, *136*, 10577; *Nat. Commun.* **2017**, *8*, 927), the present work is the **first report on atomic-level size conversion mechanism** (i.e., reaction dynamics) of metal nanoclusters (NCs). Such atomic-level reaction dynamics is revealed based on a definitive reaction equation (i.e., $[\text{Au}_{23}(\text{SR})_{16}]^+ + 2 [\text{Au}_2(\text{SR})_3]^- \rightarrow [\text{Au}_{25}(\text{SR})_{18}]^+ + 2 [\text{Au}(\text{SR})_2]^-$; SR denotes thiolate ligand), which is **for the first time** unambiguously evidenced by experiments in the present study. The as-discovered surface-motif-exchange (SME)-induced core structure transformation mechanism not only provides **original evidences on the structural relatedness** of NCs with varied core symmetries, but also highlights the decisive roles of protecting motifs on the core structure of metal NCs. In addition, the non-crystallization methodology kit demonstrated in this work could serve as **a necessary complementary means** (to X-ray crystallography) for revealing the size conversion dynamics of metal NCs. Therefore, we have strong confidence that the novelty and significance of the present study meets the requirement of *Nat. Commun.* We hope we could convince, through the detailed articulations below, that the mechanistic insights reported and the methodology developed in this manuscript will be appealing to heterogeneous readers of *Nat. Commun.* from diverse fields of noble metal chemistry, cluster chemistry, nanochemistry and materials chemistry.

1. The authors explain three major novel factors in this manuscript: a. Size conversion b. Isoelectronic conversion, c. Dynamics/kinetics by noncrystallization process;

*a. Size conversation: The size conversion of nanoclusters due to instability is very well known; it started from the 1st example of size conversion of Au_{11} to Au_{25} (*J. Am. Chem. Soc.*, **2005**, *127*, 13464–13465). Thereafter many examples are there which explains the size conversion of nanoclusters. From that perspective, this size conversation is not new.*

Reply: We agree with the reviewer that ligand exchange induced size transformation (LEIST) is well recognized in cluster research community. It has now been developed into a facile means to produce high quality thiolate-protected metal NCs based on either thiolate-protected (*Chem. Mater.* **2015**, *27*, 4289; *Chem. Mater.* **2016**, *28*, 3292; *J. Phys. Chem. Lett.* **2015**, *6*, 2976; *J. Phys. Chem. Lett.* **2015**, *6*, 2134.) or other ligands-protected (like phosphine and polymer; *J. Am. Chem. Soc.* **1997**, *119*, 12384; *J. Am. Chem. Soc.* **2005**, *127*, 13464; *J. Phys. Chem. C* **2007**, *111*, 4153) parent NCs, due to its good size monodispersity and tailorability. In a typical LEIST reaction, the size monodispersity of parent NCs could be easily transmitted into that of product NCs, while the size of product NCs could be facilely tuned by bulkiness of the incoming thiolate ligands and reaction conditions (e.g., temperature and time; *J. Phys. Chem. Lett.* **2015**, *6*, 2976). In comparison to LEIST, ligand retained size conversion reaction is less explored. As ligand retained size conversion reaction governs nucleation-growth of NPs/NCs, it has attracted increasing research interests, which have advanced the synthetic chemistry and mechanistic understandings of ligand retained size conversion reaction into *molecular level* (*J. Am. Chem. Soc.* **2014**, *136*, 10577; *Nat. Commun.* **2017**, *8*, 927). Based on such molecular-level understandings, we are now exemplifying *atomic-level reaction dynamics* of such size conversion reaction by isoelectronic conversion of $[\text{Au}_{23}(\text{SR})_{16}]^-$ into $[\text{Au}_{25}(\text{SR})_{18}]^-$. It should be pointed out that the unvaried focus of this study is the atomic-level mechanism (i.e., *behaviors of individual atoms* in clusters) governing the size conversion reaction rather than the size conversion phenomenon itself. We believe the atomic-level mechanistic insights revealed in this study not only represent an important step advancing the nanochemistry into atomic resolution, but also provide important implications to many pending fundamental puzzles in nanosynthesis, like symmetry-breaking size growth, evolution of nanocrystals, and molecular-to-metallic transition of metal materials. We also believe the non-crystallization methodology kit developed in this study could be useful to unravel reaction dynamics of other cluster reactions, prompting total resolution of the aforementioned fundamental challenges.

*b. Isoelectronic conversion: Jin et al. have recently shown such isoelectronic conversions from Au_{25} to Au_{23} for selenolates (*Chem. Mater.*, 2017, 29, 3055–3061, Ref. no 57) by addition of NaBH_4 . Then Au_{23} can be again back to Au_{25} by introducing thiols (PET) both with and without NaBH_4 case. So, such isoelectronic conversions are already reported even for same clusters, the only differences I can find of is: in this case, its solvent induced whereas for Jin et al.: its ligand and NaBH_4 induced.*

Reply: Thank you very much for your insightful comments. We might not articulate well the novelty of the present study in the original submission. Indeed, isoelectronic size conversion of $[\text{Au}_{23}(\text{SR})_{16}]^-$ to $[\text{Au}_{25}(\text{SR})_{18}]^-$ is not a new discovery. It was originally observed as a step reaction in the bottom-up growth of $[\text{Au}_{25}(\text{SR})_{18}]^-$ (*J. Am. Chem. Soc.* **2014**, *136*, 10577). As also mentioned by the reviewer, Jin and coworkers recently reported such isoelectronic conversion in selenolate (SeR)-protected Au NCs, where $[\text{Au}_{23}(\text{SeR})_{16}]^-$ could be converted into $[\text{Au}_{25}(\text{SeR})_{18-x}(\text{SR})_x]^-$ via LEIST mechanism (cited Ref. 57: *Chem. Mater.* **2017**, *29*, 3055). Although these documented explorations have advanced the mechanism study of isoelectronic size conversion to cluster- or molecular-level (i.e., molecular-like reaction equations were rationally proposed), *atomic-level insights* into such isoelectronic size conversion are lacking. We thus attempted to address this fundamental challenge in present study. This is first made possible by *unambiguously determining* the definitive reaction as $[\text{Au}_{23}(\text{SR})_{16}]^- + 2 [\text{Au}_2(\text{SR})_3]^- \rightarrow [\text{Au}_{25}(\text{SR})_{18}]^- + 2 [\text{Au}(\text{SR})_2]^-$ by mass spectrometry (MS) analyses, which are performed on both cluster and Au(I)-SR complex species with/without addition of foreign Au(I)-SR' complexes.

Based on such definitive equation, together with cluster structure features and changes reflected by the tandem MS analysis, we are then able to construct a SME-induced core structure transformation mechanism *at atomic level* for the isoelectronic size conversion reaction. Of additional note, the SME-induced core structure transformation mechanism demonstrated in this work is in sharp contrast to the reported LEIST mechanism, where the former governs the ligand retained size conversion while the latter dominates the ligand exchanged size conversion. We have revised the manuscript to convey the novelty better in the revised manuscript.

Revisions:

Page 15, Lines 3–5:

“To the best of our knowledge, this is the first definitive equation experimentally determined for size evolution reaction of thiolate-protected metal NCs.”

Page 19, Lines 2–5:

“We rationalizes that such SME-induced core structure transformation mechanism originates from the robust linear structure of SR-[Au(I)-SR]_x protecting motifs, which is in sharp contrast to the documented ligand exchange induced size transformation mechanism⁵⁰.”

c. Dynamics/kinetics by noncrystallization process: The same author group has published many of them (such as J. Am. Chem. Soc., 2014, 136, 10577–10580) for understanding the growth mechanism. Even a new article came just recently in this same journal from the same group on similar work (Nat. Commun. 8, Article number: 927, 2017) where Au₂₅ is converted to Au₄₄ (12e conversion) and the kinetics/dynamics have been studied through noncrystallization process.

Reply: Thank you for this comment. We have provided a better justification on the fundamental and technical innovations in the revised manuscript. As mentioned above, our previous work focuses on the molecular-level mechanism for the bottom-up and seeded growth of atomically precise Au NCs (*J. Am. Chem. Soc.* **2014**, 136, 10577; *Nat. Commun.* **2017**, 8, 927). In those studies, molecular-like reaction equations were proposed for evolution of Au NCs, based on the molecular formula and time-dependent abundance of intermediate [Au_n(SR)_m]^q species revealed by MS analysis. However, due to long reaction routes (consisting of dozens of elementary reactions) in those studies, we were not able to look into behavior of individual atoms (i.e., *atomic-level reaction dynamics*) in previous contributions. As such atomic-level reaction dynamics is equally important (or even more important in some senses) for completing mechanistic exploration of cluster size evolution in a broad size scale, it becomes the center of our research efforts in present work. On the basis of molecular-level understandings revealed in our previous studies, we focus on a key step reaction in bottom-up growth of [Au₂₅(SR)₁₈], i.e., [Au₂₃(SR)₁₆]⁺ → [Au₂₅(SR)₁₈]⁺, which consists of only one or two elementary reactions.

By monitoring such isoelectronic size conversion reaction in varied ionic strength, oxidizing/reducing power, and foreign Au(I)-SR' complexes by optical and MS means, we conclude that the size conversion reaction occurs via a definitive equation: [Au₂₃(SR)₁₆]⁺ + 2 [Au₂(SR)₃]⁻ → [Au₂₅(SR)₁₈]⁺ + 2 [Au(SR)₂]⁻. To the best of our knowledge, this is the *first definitive equation experimentally determined* for the size conversion of thiolate-protected metal NCs, where not only the cluster species but also the Au(I)-SR complex species involved are unambiguously analyzed by MS. More importantly, on the basis of such definitive equation together with cluster structure changes reflected by the tandem

MS measurements, we are able to construct a SME-induced core structure transformation mechanism for the isoelectronic size conversion reaction, advancing the fundamental research frontier of cluster chemistry into *atomic level*. It is demonstrated *for the first time* that the SME reaction between $[\text{Au}_2(\text{SR})_3]^-$ and $[\text{Au}(\text{SR})_2]^-$ could induce symmetry-breaking core structure transformation. Such SME-induced core structure transformation mechanism rationalizes structure diversity of metal materials at sub-3-nm regime, providing important implications to molecular-to-metallic transition of metal materials.

In addition to mechanistic insights detailed above, the non-crystallization methodology kit for revealing atomic-level size conversion reaction dynamics might also widely add to fundamental research of metal NCs and NPs. Documented fundamental research on formation and functionalization of NCs extensively relies on crystallization-based approach, where crucial intermediate NCs are crystallized and structurally examined by X-ray crystallography. As such intermediate NCs are usually co-crystallized with parent NCs, reported successes of crystallization-based approach exclusively rise from symmetry-retained ligand exchange or LEIST reactions, leaving symmetry-breaking size conversion reactions largely unexplored. Due to its nontrivial pre-detection treatment and short detection period, the as-developed MS based non-crystallization approach provides *a complementary means* for revealing atomic-level dynamics of symmetry-breaking size evolution reactions of inorganic NCs/NPs. This is especially beneficial for water-soluble NCs/NPs, as crystallization of water-soluble NCs/NPs is still challenging.

We have included the aforementioned fundamental and technical innovations in the revised manuscript.

Revisions:

Page 4, Lines 3–7:

“Based on the known NC structures and some advanced time-evolution composition/structure monitoring techniques, the size evolution mechanism of metal NCs has been investigated in a couple of contributions with molecular resolution, where the molecular-like reaction equations were proposed for the evolution of $[\text{Au}_n(\text{SR})_m]^q$.”

Page 4, Lines 21–23:

“This is most probably due to a lack of good techniques to precisely probe the composition and structure changes of clusters at atomic level.”

Page 5, Lines 1–5:

“Here, we exemplify that, beyond definitive molecular-like reaction equations, the atomic-level insights on size conversion reaction dynamics (i.e., behavior of individual atom) of metal NCs could be revealed by a noncrystallization approach based on systematic electrospray ionization mass spectrometry (ESI-MS) and tandem MS (MS/MS) analyses.”

Page 15, Lines 3–5:

“To the best of our knowledge, this is the first definitive equation experimentally determined for size evolution reaction of thiolate-protected metal NCs.”

Page 19, Lines 2–5:

“We rationalizes that such SME-induced core structure transformation mechanism originates from the robust linear structure of SR-[Au(I)-SR]_x protecting motifs, which is in sharp contrast to the documented ligand exchange induced size transformation mechanism⁵⁰.”

Page 25, Lines 1–15:

“By a combined use of steady-state and time-dependent UV-vis absorption and ESI mass spectrometry techniques to monitor both Au NC and Au(I)-SR complex species involved, we unambiguously demonstrated that the isoelectronic size conversion reaction of Au₂₃ NCs occurred via the definitive equation (1): [Au₂₃(SR)₁₆]⁻ + 2 [Au₂(SR')₃] → [Au₂₅(SR)₁₂(SR')₆]⁻ + 2 [Au(SR)₂]⁻. Based on such definitive equation, together with structural features reflected by tandem MS analysis, we constructed a SME-induced symmetry-breaking core structure transformation mechanism for the isoelectronic size conversion reaction, experimentally corroborating the structural relatedness of Au NCs with varied core symmetry. The SME-induced symmetry-breaking core structure transformation mechanism thus not only rationalizes the structure diversity of metal NCs in sub-3 nm regime, but also offers insightful clues towards demystification of long-standing fundamental puzzles, such as nucleation-growth of nanocrystals, molecular-to-metallic transition, and emergence of collective physicochemical properties (e.g., surface plasmon resonance) of metal materials.”

Other minor issues:

2. *Is that conversion is very specific about ligands? Only p-MBA shows such conversion? This Au₂₃ cluster can also be synthesized with a variety of ligands. Will this solvent induced conversion also work for them?*

Reply: We appreciate the insightful comment and suggestion from the reviewer, which inspired us to examine the ligand effects on the size conversion reaction. In addition to *para*-mercaptobenzoic acid (*p*-MBA), we also prepared *m*-MBA and *o*-MBA protected Au₂₃ NCs by optimizing the solvent polarity (volume fraction of ethanol in its mixture with water, *f*_{EtOH}) and pH. As shown in Fig. RL-1 (black lines), the as-prepared [Au₂₃(*m*-MBA)₁₆]⁻ (*f*_{EtOH} = 0.4 and pH = 12.3) and [Au₂₃(*o*-MBA)₁₆]⁻ (*f*_{EtOH} = 0.2 and pH = 10.0) exhibit dominant peaks at ~590 nm in ultraviolet-visible (UV-vis) absorption spectra, similar to that of [Au₂₃(*p*-MBA)₁₆]⁻ (*f*_{EtOH} = 0.4 and pH = 12.3; Fig. 1b). Incubation of [Au₂₃(*m*-MBA)₁₆]⁻ in pure water for 2 days could diminish characteristic peaks of [Au₂₃(SR)₁₆]⁻ (Fig. RL-1a), giving rise to characteristic peaks of [Au₂₅(SR)₁₈]⁻ (e.g., dominant peak at 690 nm). Similar solvent-induced size conversion from [Au₂₃(SR)₁₆]⁻ to [Au₂₅(SR)₁₈]⁻ is however not observed for [Au₂₃(*o*-MBA)₁₆]⁻. As can be seen in Fig. RL-1b, incubation of [Au₂₃(*o*-MBA)₁₆]⁻ could only cause slight intensity decrease of the characteristic peaks of [Au₂₃(SR)₁₆]⁻. A similar solvent-induced size conversion was not found in organic-soluble [Au₂₃(SR)₁₆]⁻ NCs. In a particular note, Jin and coworkers prepared [Au₂₃(S-*c*-C₆H₁₁)₁₆]⁻, where HS-*c*-C₆H₁₁ denoted 1-cyclohexanethiol, in water/methanol (4/10, Vol/Vol) and crystallized them in dichloromethane (*J. Am. Chem. Soc.* **2013**, *135*, 18264). Such switch from synthesis to crystallization solvents did not cause significant degradation or size conversion of [Au₂₃(S-*c*-C₆H₁₁)₁₆]⁻. Taken together, the solvent-induced size conversion of [Au₂₃(SR)₁₆]⁻ could be fairly extended to thiolates (e.g., *m*-MBA) structurally close to *p*-MBA. For other thiolate ligands (e.g., *o*-MBA and -S-C₆H₁₁), this approach needs to be further experimentally verified in a broader spectrum of

solvents. We believe that the successfulness of such solvent-induced size conversion approach is largely dependent on the affinity of SR-[Au(I)-SR]₃ protecting motifs of [Au₂₃(SR)₁₆]⁻ to the incubating solvent.

Figure RL-1. Ultraviolet-visible absorption spectra of (a) [Au₂₃(*m*-MBA)₁₆]⁻ and (b) [Au₂₃(*o*-MBA)₁₆]⁻ before (black lines) and after (magenta lines) incubation in pure water for 2 days, where *m*-MBA and *o*-MBA are *meta*-mercaptobenzoic acid and *ortho*-mercaptobenzoic acid, respectively. The dominant absorption features of NCs are indicated by arrows in each spectrum.

3. In Figure 2: The author says “that the reaction kinetics is rarely dependent on the chemical identities of anions, but it is highly related to the ionic strength”. If the reaction kinetics is pseudo 1st order then how ionic strength effects in the size conversion rate kinetics? More explanation is needed here.

Reply: Thank you for this good suggestion. The size conversion reaction depends on the collision of negatively charged [Au₂₃(SR)₁₆]⁻ and similarly negatively charged [Au₂(SR)₃]⁻, where the negative surface charge of the former participates not only from its net -1 charge but also from deprotonation of surface *p*-MBA ligands. Therefore, the double layer developed from such high-density surface charge of [Au₂₃(SR)₁₆]⁻ could be effectively squeezed at high ionic strength, facilitating the collision and reaction of [Au₂₃(SR)₁₆]⁻ with [Au₂(SR)₃]⁻. We have revised the manuscript according to the reviewer’s good suggestion.

Revisions:

Page 12, Line 21 – Page 13, Line 4:

“The as-demonstrated ionic strength dependence should be attributed to electrostatic repulsion between the similarly negatively-charged [Au₂₃(SR)₁₆]⁻ and Au(I)-SR complex species, of which [Au(SR)₂]⁻, [Au₂(SR)₃]⁻, [Au₃(SR)₄]⁻, and [Au₄(SR)₅]⁻ could be the candidates. At high ionic strength, the squeezed double layer of [Au₂₃(SR)₁₆]⁻ could weaken its electrostatic repulsion with negatively-charged Au(I)-SR complex species, facilitating their collision and thus size conversion reaction.”

4. Figure 3: Each stack spectral intensity axis can be narrow down as there is more free space available between each stack. This will help to visualize the spectra in a better way.

Reply: Thank you for your good suggestion. In the revised manuscript, the y-axis in Fig. 3 has been optimized for an easy and clear reading.

Revisions:

Page 13, Figure 3:

The y-axis in each panel has been optimized.

Page 13, Caption of Figure 3:

“Electrospray ionization mass spectra of $[\text{Au}_{25}(\text{SR})_{18}]^-$ nanoclusters formed by reacting $[\text{Au}_{23}(p\text{-MBA})_{16}]^-$ with varied dosage of Au(I)-(*p*-NTP) complexes, which are expressed by $[\text{Au}(\text{I})] = (\mathbf{a}, \mathbf{e}, \mathbf{i})$ 0 mM, $(\mathbf{b}, \mathbf{f}, \mathbf{j})$ 0.1 mM, $(\mathbf{c}, \mathbf{g}, \mathbf{k})$ 0.5 mM and $(\mathbf{d}, \mathbf{h}, \mathbf{l})$ 1.0 mM. $(\mathbf{a-d})$ Broad range spectra at $m/z = 1000\text{--}4000$; $(\mathbf{e-h})$ zoom-in spectra of cluster peaks carrying 4- charges; and $(\mathbf{i-l})$ experimental (black lines) and simulated (magenta lines) isotope patterns of $[\text{Au}_{25}(p\text{-MBA})_{18-x}(p\text{-NTP})_x - 3\text{H}]^{4-}$ with x values indicated in the corresponding panels.”

5. Ref. 9. does not fit there; a recent important review article on nanoclusters (*Chem. Rev.*, 2017, 117 (12), 8208–8271) should replace ref. 9.

Reply: Thank you for this good suggestion. We have included this nice review article accordingly.

Revisions:

Page 3, Reference 9:

The Ref. 9 has been corrected as *Chem. Rev.*, 2017, 117, 8208.

6. In all the absorbance based figures (such as Figure 1 b,c; 2a, etc.), absorbance value is not there. Absorbance should always have some value, it cannot be arbitrary. If you have normalized, then mention normalized absorbance.

Reply: The normalized absorbance has been indicated on the y-axis of all UV-vis absorption spectra. Thank you.

Revisions:

Page 28, Lines 15–18:

“UV-vis absorption spectra were obtained on a Shimadzu UV-1800 spectrometer, and the optical intensity was normalized to the characteristic absorption at 589 nm of fresh $[\text{Au}_{23}(\text{SR})_{16}]^-$ used in size conversion exploration unless otherwise indicated.”

Figures 1, 2, 6, and Supplementary Figures 2–5, 7 and 8:

The normalized absorbance values have been added to UV-vis absorption spectra.

7. Many times, the calculated mass spectra and experimental mass spectra do not properly match (such as Figure 3c, $x = 6, 9, 12$, SI Figure 2b, SI Figure 3a,) whereas others such as SI Figure 9 to 12 matches perfectly. Anything wrong in the assignment for the earlier case? Figure S13 would be better to consider as noise unless one has better spectra.

Reply: Thank you for your insightful suggestions. The mass and molecular formula of clusters are deduced according to the most intensive isotope peak in the high-resolution electrospray ionization (ESI) mass spectra, which is a common practice in cluster literature (*J. Am. Chem. Soc.* **2015**, *137*, 11578; *J. Am. Chem. Soc.* **2016**, *138*, 14727; *Chem. Mater.* **2016**, *28*, 3292; *J. Phys. Chem. Lett.* **2012**, *3*, 1997; *Nat. Commun.* **2016**, *7*, 13447; *J. Am. Chem. Soc.* **2015**, *137*, 1206; *J. Am. Chem. Soc.* **2014**, *136*, 17016). We agree with the reviewer that the experimental and simulated isotope patterns in the as-mentioned mass spectra, i.e., Fig. 3, Supplementary Fig. 2b and 3b, exhibit relatively larger mismatches. However, we would like to point out that all observable mismatches in these spectra are within 0.5 Da, which suggests the good accuracy of our assignment. It should be mentioned that similar or even larger mismatches are commonly found in the reported mass spectra of atomically precise metal NCs (*Nat. Commun.* **2015**, *6*, 8667; *J. Am. Chem. Soc.* **2015**, *137*, 1206; *J. Am. Chem. Soc.* **2014**, *136*, 17016; *J. Phys. Chem. Lett.* **2014**, *5*, 3757).

This insightful comment also motivated us to seek the root cause of such mismatches. We first re-examined the assignment of Fig. 3, spotting extra isotope peaks prior to the simulated ones as the most notable mismatch. For example, three extra isotope peaks could be identified prior to the simulated isotope pattern of $[\text{Au}_{25}(\text{p-MBA})_{12}(\text{p-NTP})_6 - 3 \text{H}]^4$ in Fig. 3j, a zoom-in view of which is included as Fig. RL-2 below. Such extra peaks (Peaks 1-3 in Fig. RL-2) could be attributed to $[\text{Au}_{25}(\text{p-MBA})_{15}(\text{p-NTP})_3 - 3 \text{H}]^4$ (blue line in Fig. RL-2), which is a side product by substituting one molecule of the incoming $[\text{Au}_2(\text{p-NTP})_3]^-$ by the residual $[\text{Au}_2(\text{p-MBA})_3]^-$ in the size conversion reaction detailed in equation (1). More importantly, the perfect match in intensity profile between such experimental (Peaks 1-3 in Fig. RL-2) and simulated (blue line in Fig. RL-2) peaks has ruled out any possible presence of $[\text{Au}_{25}(\text{p-MBA})_{18-x}(\text{p-NTP})_x]^-$ ($x = 1$ or 2) species, agreeing well with the proposed SR- $[\text{Au}(\text{I})\text{-SR}]_2$ based SME mechanism. It should be reminded that the size conversion from $[\text{Au}_{23}(\text{p-MBA})_{16}]^-$ to $[\text{Au}_{25}(\text{p-MBA})_{18}]^-$ could be induced by the residual Au(I)-(p-MBA) complexes in the absence of foreign Au(I)-(p-NTP) complexes (Fig. 1). As the residual Au(I)-(p-MBA) complexes could not be completely eliminated from the $[\text{Au}_{23}(\text{p-MBA})_{16}]^-$ solution by ultrafiltration (the optimal technique for purifying $[\text{Au}_{23}(\text{p-MBA})_{16}]^-$ NCs, while other purification means could significantly change the cluster quality), it is reasonable to expect $[\text{Au}_{25}(\text{p-MBA})_{15}(\text{p-NTP})_3]^-$ as a side product in Au(I)-(p-NTP) complexes induced size conversion reaction. Similarly, the experimental isotope patterns in Fig. 3k and 3l could be perfectly deconvoluted to $[\text{Au}_{25}(\text{p-MBA})_{18-x}(\text{p-NTP})_x]^-$ ($x = 6, 9, \text{ and } 12$; Fig. RL-3), where $[\text{Au}_{25}(\text{p-MBA})_9(\text{p-NTP})_9]^-$ and $[\text{Au}_{25}(\text{p-MBA})_6(\text{p-NTP})_{12}]^-$ are dominant products, respectively.

Figure RL-2. Zoom-in view of experimental (black line) and simulated (colored lines) isotope patterns of $[\text{Au}_{25}(\text{p-MBA})_{18-x}(\text{p-NTP})_x]^-$ clusters formed by reacting $[\text{Au}_{23}(\text{p-MBA})_{16}]^-$ with the $\text{Au(I)}-(\text{p-NTP})$ complexes ($[\text{Au(I)}] = 0.1 \text{ mM}$), where p-MBA and p-NTP are *para*-mercaptobenzoic acid and *para*-nitrothiophenol, respectively. Color code: blue, $x = 3$; magenta, $x = 6$.

Figure RL-3. Experimental (black lines) and simulated (colored lines) isotope patterns of $[\text{Au}_{25}(\text{p-MBA})_{18-x}(\text{p-NTP})_x]^-$ clusters formed by reacting $[\text{Au}_{23}(\text{p-MBA})_{16}]^-$ with varied dosage of $\text{Au(I)}-(\text{p-NTP})$ complexes (expressed by $[\text{Au(I)}]$ in each spectrum), where p-MBA and p-NTP are *para*-mercaptobenzoic acid and *para*-nitrothiophenol, respectively. Color code: olive, $x = 0$; blue, $x = 3$; magenta, $x = 6$; cyan, $x = 9$; wine, $x = 12$.

Different from Fig. 3, the mismatch observed in Supplementary Fig. 2b and 3b is <0.5 Da mass shift of the experimental isotope patterns. In comparison to those reported in Supplementary Fig. 10-20, this relatively larger mass shift could be attributed to the equipment error caused by varied testing media. Supplementary Fig. 2b and 3b were recorded on organic solutions of phase-transferred Au NCs, while Supplementary Fig. 10-20 were recorded on aqueous solutions of Au NCs. Of note, the unvaried core size of $[\text{Au}_{23}(\textit{p}\text{-MBA})_{16}]^-$ NCs after phase-transfer is also evidenced by their UV-vis absorption spectra (Supplementary Fig. 2a and 3a), showing characteristic absorption features of pure $[\text{Au}_{23}(\text{SR})_{16}]^-$ NCs. Regarding to Supplementary Fig. 14 (Supplementary Fig. 13 in the original submission), it would be better attributed to fragment cluster ions of $[\text{Au}_{21}(\text{SR})_{14} + x \text{Na} - (x + 4) \text{H}]^{3-}$ rather than noise. This is because the regular inter-peak spacing of $m/z = 7.33$ in the top panel of Supplementary Fig. 14 could be easily transmitted to a mass difference of 22 ($= 7.33 \times 3$), corresponding to successive Na^+ coordination and H^+ dissociation (i.e., $+ \text{Na} - \text{H}$) by cluster ions. Similar regularly-spaced peak groves are also observed in mass spectra of other cluster ions (e.g., Supplementary Fig. 10-13 and 15-20), supporting our assignment of Supplementary Fig. 14.

Revisions:

Page 14, Lines 12–23:

“The most intriguing finding is stepwise (with a pace of 3 Da) mass shift when the dosage of Au(I)-(*p*-NTP) complexes is increased (Supplementary Fig. 9 and Supplementary Note 3). Remarkably, in comparison to the reference $[\text{Au}_{25}(\textit{p}\text{-MBA})_{18}]^-$ peak (Fig. 3a, 3e and 3i), the most intensive isotope peak exhibits a mass increment of initially 6 Da (Fig. 3b, 3f and 3j) and thereafter 3 Da (Fig. 3c, 3d, 3g, 3h, 3k and 3l), corresponding to substitutions of 6 and 3 *p*-MBA by *p*-NTP, respectively. Considering similar linear configuration (i.e., $\text{SR}-[\text{Au}(\text{I})\text{-SR}]_n$) of Au(I)-SR complexes and Au(I)-SR protecting motifs, together with the centrosymmetry of $[\text{Au}_{23}(\text{SR})_{16}]^-$, such stepwise ligand substitution suggests a $\text{SR}-[\text{Au}(\text{I})\text{-SR}]_2$ motif based association mechanism. As a dominant pathway, the size conversion reaction of Au_{23} to Au_{25} NCs is induced by association of 2 molecules of (*p*-NTP)- $[\text{Au}(\text{I})\text{-}(\textit{p}\text{-NTP})]_2$ with $[\text{Au}_{23}(\textit{p}\text{-MBA})_{16}]^-$ at a low dosage of complexes ($[\text{Au}(\text{I})] = 0.1 \text{ mM}$).”

Supplementary Information (SI), Pages 11–12, Supplementary Note 3:

“The ESI mass spectra reported in Fig. 3 suggest a $\text{SR}-[\text{Au}(\text{I})\text{-SR}]_2$ based surface-motif-exchange (SME)-induced size conversion reaction. Such mechanism could first be understood by comparing the most intensive isotope peaks. As shown in Fig. 3i-3l, positive mass shifts of initially 6 Da ($= 2 \times 3 \text{ Da}$, Fig. 3j) and subsequently 3 Da (Fig. 3k and 3l) from the reference $[\text{Au}_{25}(\textit{p}\text{-MBA})_{18}]^-$ peak (*p*-MBA = *para*-mercaptobenzoic acid) could be observed with increasing dosage of Au(I)-(*p*-NTP) complexes (*p*-NTP = *para*-nitrothiophenol). Given the molecular weight difference of the incoming *p*-NTP and *p*-MBA ligands to be 1 Da, such 3 Da based mass shift suggests the size conversion occurs by association of (*p*-NTP)- $[\text{Au}(\text{I})\text{-}(\textit{p}\text{-NTP})]_2$ to $[\text{Au}_{23}(\textit{p}\text{-MBA})_{16}]^-$. Taking the centrosymmetry of $[\text{Au}_{23}(\text{SR})_{16}]^-$ into account, the initial 6 Da mass shift suggests the dominant size conversion pathway as equation (S2) below.

In order to fully understand the size conversion reaction at molecular level, a minor discrepancy between experimental and simulated isotope patterns in Fig. 3i-3l should not be neglected. For example, three extra isotope peaks could be identified prior to the simulated isotope pattern of $[\text{Au}_{25}(\textit{p}\text{-MBA})_{12}(\textit{p}\text{-NTP})_6 - 3 \text{H}]^{4+}$ in the experimental spectrum of Fig. 3j. Such extra peaks (Peaks

1-3 in Supplementary Fig. 9) could nevertheless be assigned to a side product of $[\text{Au}_{25}(\text{p-MBA})_{15}(\text{p-NTP})_3 - 3 \text{H}]^{4+}$ (blue lines in Supplementary Fig. 9). More importantly, the perfect intensity profile match between such experimental (Peaks 1-3 in Supplementary Fig. 9) and simulated (blue lines in Supplementary Fig. 9) peaks rules out any possible presence of $[\text{Au}_{25}(\text{p-MBA})_{18-x}(\text{p-NTP})_x]^{-}$ ($x = 1$ or 2) species, agreeing perfectly with the proposed SR- $[\text{Au}(\text{I})\text{-SR}]_2$ based SME mechanism. It should be reminded that the size conversion from $[\text{Au}_{23}(\text{p-MBA})_{16}]^{-}$ to $[\text{Au}_{25}(\text{p-MBA})_{18}]^{-}$ could be induced by the residual Au(I)-(*p*-MBA) complexes in the absence of the foreign Au(I)-(*p*-NTP) complexes (Fig. 1). Substituting one molecule of $[\text{Au}_2(\text{p-NTP})_3]^{-}$ by the residual $[\text{Au}_2(\text{p-MBA})_3]^{-}$ in equation (S2) could yield $[\text{Au}_{25}(\text{p-MBA})_{15}(\text{p-NTP})_3 - 3 \text{H}]^{4+}$. Similarly, the experimental isotope patterns in Fig. 3k and 3l could be perfectly deconvoluted to $[\text{Au}_{25}(\text{p-MBA})_{18-x}(\text{p-NTP})_x]^{-}$ ($x = 3, 6, 9,$ and 12 ; Supplementary Fig. 9), where $[\text{Au}_{25}(\text{p-MBA})_9(\text{p-NTP})_9]^{-}$ and $[\text{Au}_{25}(\text{p-MBA})_6(\text{p-NTP})_{12}]^{-}$ are dominant products in the SME reactions with 0.5 and 1.0 mM Au(I)-(*p*-NTP) complexes, respectively. The as-evidenced stepwise mass increment by 3 Da unambiguously indicates the SR- $[\text{Au}(\text{I})\text{-SR}]_2$ motif as the structure basis for the size conversion reaction.”

SI, Page 13, Supplementary Figure 9:

Fig. RL-3 is included as Supplementary Fig. 9.

Comments by Reviewer #2:

Ligand protected metal clusters have attracted much attention due to their unique physical and chemical properties. Various characterization techniques have progressed for these clusters, much knowledge has been accumulated. One of them is mass spectrometry, which has made it possible to handle metal clusters as molecule or aggregate of atoms. Recently, X-ray crystal structure analysis has become essential techniques to understand synthesized clusters. However, mass spectrometry is also important technique yet. In this work, the authors explored the synthesis mechanism from $[Au_{23}(SR)_{16}]^-$ to $[Au_{25}(SR)_{18}]^-$ using ESI mass spectrometry. They take advantage of characteristics of mass spectrometry in this work. Additionally, not only many experiments are conducted, but also well analyzed about obtained results. this knowledge and information will fascinate readers of various fields. Therefore, I think this work deserve potential for publication to Nature Communications. To become this work better, I have some minor comments and questions. I am very glad if you will consider about these.

Reply: We are glad that the reviewer finds this work interesting and informative. Indeed, since several pioneered works on MS of metal NCs (*J. Am. Chem. Soc.* **2005**, *127*, 5261; *J. Phys. Chem. B* **1998**, *102*, 10643), MS has been developed into an indispensable tool for size and structure analyses of metal NCs. The demystification of isoelectronic size conversion mechanism at unprecedented molecular level (reaction kinetics) and atomic level (reaction dynamics) in this study is largely made possible by delicate MS (including MS/MS) analyses. Based on this MS-based noncrystallization approach, we exemplify that the symmetry-breaking size conversion from $[Au_{23}(SR)_{16}]^-$ to $[Au_{25}(SR)_{18}]^-$ occurs via a SME-induced core structure transformation mechanism, which would surface as a definitive equation of $[Au_{23}(SR)_{16}]^- + 2 [Au_2(SR)_3]^- \rightarrow [Au_{25}(SR)_{18}]^- + 2 [Au(SR)_2]^-$. The as-revealed SME-induced core structure transformation mechanism not only highlights the critical role of surface motifs in dictating the size and structure of metal NCs, but also rationalizes the structure diversity of thiolate-protected metal NCs at sub-3 nm regime. We are also in complete agreement with the reviewer that the reported MS-based methodology kit is an indispensable alternative to X-ray crystallography, conducive for precisely understanding the growth and size conversion mechanism of metal NCs. This would be especially beneficial for aqueous and symmetry-breaking NC reactions, as crystallization of intermediate NCs in these reactions still remains as one of the greatest challenges at current stage of cluster research. Last but not least, we would like to gratefully appreciate the reviewers' insightful and encouraging comments/suggestions, which have been taken into careful account in this revision (see our detailed responses and revisions below).

1. In page 9, the authors explain why the conversion occurs from $[Au_{23}(SR)_{16}]^-$ to $[Au_{25}(SR)_{18}]^-$ by changing a solvent. If $[Au_{23}(SR)_{16}]^-$ prefers to have long oligomer motif, $[Au_{23}(SR)_{16}]^-$ seems to decompose instead of conversion to $[Au_{25}(SR)_{18}]^-$? Why did not this cluster decompose?

Reply: Indeed, the documented X-ray crystallography data has revealed that $[Au_{23}(SR)_{16}]^-$ features with two long SR-[Au(I)-SR]₃ motifs, which hold the three layers of its cuboctahedral Au₁₃ core together (*J. Am. Chem. Soc.* **2013**, *135*, 18264). The critical role of such long SR-[Au(I)-SR]₃ motifs on stabilizing the cuboctahedral Au₁₃ core has also been evidenced by a series of ligand exchange induced size transformation explorations (*Sci. Adv.* 2017, **3**, e1603193; *J. Phys. Chem. Lett.* **2017**, **8**, 866). It has been extensively documented that quasi-FCC symmetry (cuboctahedron could be regarded as a fragment of FCC lattice) could be retained in the metal core of product NCs, as long as the trimeric SR-[Au(I)-SR]₃

motifs of $[\text{Au}_{23}(\text{SR})_{16}]^-$ are kept complete. The symmetry-breaking (cuboctahedral-to-icosahedral core) size conversion reaction in this study is actually made possible by disrupting such trimeric $\text{SR}-[\text{Au}(\text{I})-\text{SR}]_3$ motifs. Since increasing the solvent polarity favors shorter water-soluble $\text{SR}-[\text{Au}(\text{I})-\text{SR}]_x$ motifs, adjusting the solvent from water/ethanol (6/4, Vol/Vol) to pure water in the present study could effectively “cut” the $\text{SR}-[\text{Au}(\text{I})-\text{SR}]_3$ motifs into shorter analogs, of which $\text{SR}-[\text{Au}(\text{I})-\text{SR}]_2$ is a good candidate. The destruction of $\text{SR}-[\text{Au}(\text{I})-\text{SR}]_3$ motifs would then allow structure transformation of the Au_{13} core from cuboctahedron to icosahedron. As a net result, changing the solvent from water/ethanol (6/4, Vol/Vol) to pure water could drive the size conversion from $[\text{Au}_{23}(\text{SR})_{16}]^-$ to more stable $[\text{Au}_{25}(\text{SR})_{18}]^-$, which features an icosahedral Au_{13} core capped by 6 $\text{SR}-[\text{Au}(\text{I})-\text{SR}]_2$ motifs.

We agree with the reviewer that another possible reaction route of $[\text{Au}_{23}(\text{SR})_{16}]^-$ is decomposition into $\text{Au}(\text{I})-\text{SR}$ complexes, as $\text{Au}(\text{I})-\text{SR}$ complexes is thermodynamically more stable than $[\text{Au}_{23}(\text{SR})_{16}]^-$. However, such decomposition reaction would preferentially occur in the presence of excess free thiols and at elevated temperature (*Small* **2007**, 3, 835). Therefore, the current reaction conditions (by simply changing the solvent) are more likely to induce size conversion reaction rather than the decomposition reaction.

2. In page 16, association between 2 $\text{SR}-[\text{Au}(\text{I})\text{SR}]_2$ and $[\text{Au}_{23}(\text{SR})_{16}]^-$ is describe. Where does 2 $\text{SR}-[\text{Au}(\text{I})\text{SR}]_2$ come from?

Reply: $\text{SR}-[\text{Au}(\text{I})-\text{SR}]_2$ is sourced from the $\text{Au}(\text{I})-\text{SR}$ complexes used in the synthesis of $[\text{Au}_{23}(\text{SR})_{16}]^-$. $[\text{Au}_{23}(\text{SR})_{16}]^-$ was synthesized by CO-reduction of $\text{Au}(\text{I})-\text{SR}$ complexes, and $\text{SR}-[\text{Au}(\text{I})-\text{SR}]_2$ could be a residue of such $\text{Au}(\text{I})-\text{SR}$ complexes. Although we purified the $[\text{Au}_{23}(\text{SR})_{16}]^-$ by ultrafiltration prior to size conversion exploration, such ultrafiltration process was not capable of completely removing $\text{SR}-[\text{Au}(\text{I})-\text{SR}]_2$. The presence of $\text{SR}-[\text{Au}(\text{I})-\text{SR}]_2$ in the purified $[\text{Au}_{23}(\text{SR})_{16}]^-$ used for size conversion exploration is experimentally supported by our MS analysis (Supplementary Fig. 6).

3. In page 23, the high stability of $[\text{Au}_{23}(\text{SR})_{16}@x\text{CTA}]^-$ was introduced. But only a time course of this optical absorption spectrum was shown in Figure 6. You might compare this result to that of $[\text{Au}_{23}(\text{SR})_{16}]^-$ in water/ethanol. By adding this information, reader can understand the difference of stability easily.

Reply: Thank you again for your good suggestion, which has encouraged us to compare the degradation profiles of $[\text{Au}_{23}(\text{SR})_{16}]^-$ with phase-transferred $[\text{Au}_{23}(\text{SR})_{16}@x\text{CTA}]^-$. As can be seen from Fig. RL-4, the degradation rate of $[\text{Au}_{23}(\text{SR})_{16}]^-$ in either water or the simulated mother liquid (water/ethanol, 6/4, Vol/Vol) is distinctively higher than that of $[\text{Au}_{23}(\text{SR})_{16}@x\text{CTA}]^-$, providing supportive evidence of the superior stability of the latter. We have included Fig. RL-4 in this revision as Supplementary Fig. 21.

Revision:

Page 24, Lines 3–7:

“As shown in the time-course UV-vis absorption spectra (Fig. 6), an elongated incubation of $[\text{Au}_{23}(\text{SR})_{16}@x\text{CTA}]^-$ for 4 weeks only led to a slight change of their characteristic absorption peaks, which is in sharp contrast to an accelerated degradation of $[\text{Au}_{23}(\text{SR})_{16}]^-$ in either water or the simulated mother liquid (water/ethanol, 6/4, Vol/Vol) within 2 days (Supplementary Fig. 21).”

SI, Page 25, Supplementary Figure 21:

Fig. RL-4 has been included as Supplementary Fig. 21.

Figure RL-4. Time-course absorbance of the characteristic peak at 589 nm of $[\text{Au}_{23}(\text{SR})_{16}]^-$ and $[\text{Au}_{23}(\text{SR})_{16}@x\text{CTA}]^-$ NCs, where SR and CTA denote thiolate and cetyltrimethylammonium ligands, respectively. $[\text{Au}_{23}(\text{SR})_{16}]^-$ NCs are dissolved in water and the simulated mother liquid (water/ethanol = 6/4, Vol/Vol), while $[\text{Au}_{23}(\text{SR})_{16}@x\text{CTA}]^-$ NCs are dissolved in ethanol. The original time-course ultraviolet-visible absorption spectra could be found in Fig. 2a, 6 and Supplementary Fig. 4, respectively.

4. In this work, the conversion from $[\text{Au}_{23}(\text{SR})_{16}]^-$ to $[\text{Au}_{25}(\text{SR})_{18}]^-$ occurs by changing a solvent from water/ethanol to pure water. Is this reaction reversible?

Reply: This is another very inspiring comment. As suggested by the reviewer, we examined the reversibility of the solvent-induced size conversion reaction. In particular, water-soluble $[\text{Au}_{25}(p\text{-MBA})_{18}]^-$ NCs were incubated in a mixture of water and ethanol (water/ethanol, 6/4, Vol/Vol) for 2 days. UV-vis absorption spectrometry analysis (Fig. RL-5) suggests that such incubation could negligibly change the purity of $[\text{Au}_{25}(\text{SR})_{18}]^-$. This observation readily suggests that the solvent-induced size conversion reaction of $[\text{Au}_{23}(\text{SR})_{16}]^- \rightarrow [\text{Au}_{25}(\text{SR})_{18}]^-$ is irreversible, which should be attributed to the remarkable stability of $[\text{Au}_{25}(\text{SR})_{18}]^-$ in solution.

Figure RL-5. UV-vis absorption spectra of fresh (black line) and incubated (for 2 days, magenta line) $[\text{Au}_{25}(\text{SR})_{18}]^{-}$ NCs. SR denotes thiolate ligand. The absorbance is normalized to the characteristic absorbance of fresh $[\text{Au}_{25}(\text{SR})_{18}]^{-}$ NCs at 690 nm.

5. *Experimental isotope patterns in supplementary figure 2 and 3 do not seem to fit calculation data. Is this of calibration problem?*

Reply: We would like to thank the reviewer for his/her careful eyes! We agree with the reviewer that the experimental and simulated isotope patterns in Supplementary Fig. 2b and 3b exhibit relatively larger mismatch. However, we would like to point out that the observable mismatches in these spectra are within 0.5 Da, which again suggests the good accuracy of our assignment. Similar or even larger mismatches are commonly found in the reported mass spectra of atomically precise metal NCs (*Nat. Commun.* **2015**, *6*, 8667; *J. Am. Chem. Soc.* **2015**, *137*, 1206; *J. Am. Chem. Soc.* **2014**, *136*, 17016; *J. Phys. Chem. Lett.* **2014**, *5*, 3757). Of note, the unvaried core size of $[\text{Au}_{23}(p\text{-MBA})_{16}]^{-}$ NCs after phase-transfer is also evidenced by their UV-vis absorption spectra (Supplementary Fig. 2a and 3a), showing the characteristic absorption features of pure $[\text{Au}_{23}(\text{SR})_{16}]^{-}$ NCs. We agree with the reviewer that the relatively larger mismatch in Supplementary Fig. 2b and 3b could be attributed to the equipment due to the changes in the operating media for the phase-transferred Au NCs.

REVIEWERS' COMMENTS:

Reviewer #1 (Remarks to the Author):

The revised version has improved well to be suitable for this journal. Some issues need to be addressed before its getting accepted.

1. In abstract: Throughout the abstract, the author demonstrates the nanoclusters as nanoparticles (NPs) which would be better to correct as 'nanoclusters' (NCs) (e.g.: line 20, 24, etc.).
2. Make a list of contents in SI (at the very first page of SI)
3. Figure S4 and few other SI figures: The author says, normalized absorbance but are these spectra really normalized? If so, which peak position was chosen to normalize them? In response, the author says 589 peak was chosen for normalization. If that so, then all the kinetics data are completely invalid; For kinetics measurement, you should not normalize the absorption spectra which will mislead the information as absorbance ratio is the key point to monitor such kinetics. In the last review, it was suggested to give appropriate absorbance values or write 'normalized if you have done normalization'. That does not mean, in all spectra, you can just write "normalized absorbance" blindly. Even though it appears a small error but it can be a very big basic mistake. Please take care of this.

Reviewer #2 (Remarks to the Author):

I am really glad about you reply. Now, all of my dubious points are solved, thereby I think that the quality of this manuscript improved. Thus, I can suggest the acceptance of this manuscript to Nature Communications.

Replies to reviewers' comments and descriptions of revisions made

Comments by Reviewer #1:

The revised version has improved well to be suitable for this journal. Some issues need to be addressed before its getting accepted.

Reply: We are glad that the reviewer finds improvements in the revised manuscript, many of which are spurred from the useful comments and suggestions from the reviewer. We would also like to thank the reviewer for review and comments on our manuscript again. All remaining concerns have been taken into careful consideration in this revision, and a point-to-point response to the specific comments could be found in the coming paragraphs below.

1. In abstract: Throughout the abstract, the author demonstrates the nanoclusters as nanoparticles (NPs) which would be better to correct as 'nanoclusters' (NCs) (e.g.: line 20, 24, etc.).

Reply: The miscellaneous use of the terms “NPs” and “NCs” has been avoided throughout the manuscript. Thank you for your good suggestion!

Revisions:

Page 2, Abstract:

The term “nanoclusters (NCs)” has been defined and replaced “NPs” whereas applicable.

2. Make a list of contents in SI (at the very first page of SI)

Reply: We agree with the reviewer that a Table of Content page could be conducive for an easy reading of Supplementary Information. However, such Table of Content page is currently not permitted by the format requirements of *Nature Communications*.

3. Figure S4 and few other SI figures: The author says, normalized absorbance but are these spectra really normalized? If so, which peak position was chosen to normalize them? In response, the author says 589 peak was chosen for normalization. If that so, then all the kinetics data are completely invalid; For kinetics measurement, you should not normalize the absorption spectra which will mislead the information as absorbance ratio is the key point to monitor such kinetics. In the last review, it was suggested to give appropriate absorbance values or write 'normalized if you have done normalization'. That does not mean, in all spectra, you can just write “normalized absorbance” blindly. Even though it appears a small error but it can be a very big basic mistake. Please take care of this.

Reply: We appreciate the reviewer's careful and rigorous attitude towards scientific presentation of data. The absorbance of characteristic peak at 589 nm was taken as a proxy of the concentration of [Au₂₃(SR)₁₆] in the kinetic analyses. In individual kinetic analysis, the time-dependent optical density at

589 nm ($OD'_{@589}$) was recorded and normalized to that at $t = 0$ ($OD'_{@589,0}$) by the equation $OD_{@589} = OD'_{@589}/OD'_{@589,0}$, where $OD_{@589}$ denotes normalized optical density at 589 nm. Such normalization could not cause any interference to the optical density ratio, and therefore will not affect the result of kinetic analysis. To eliminate any possible confusion, the normalization method has been clearly stated in figure captions whereas applicable.

Revisions:

Captions of Figures 2, 6 and Supplementary Figure 4:

“All absorption spectra are normalized to optical density at 589 nm at $t = 0$.”

Captions of Supplementary Figures 5, 7, 8 and 21:

“Time-course absorption spectra in individual kinetic analysis are normalized to optical density at 589 nm at $t = 0$.”

Comments by Reviewer #2:

I am really glad about your reply. Now, all of my dubious points are solved, thereby I think that the quality of this manuscript improved. Thus, I can suggest the acceptance of this manuscript to Nature Communications.

Reply: We are very glad to learn that our revisions are satisfactory to the reviewer. We would like to thank the reviewer again for his/her constructive comments and suggestions, which have spurred significant improvements on both the scientific content and readability of this manuscript.